# Development of summary indices of antenatal care service quality in Haiti, Malawi and Tanzania

Ashley Sheffel [ID],[1] Scott Zeger,[1,2] Rebecca Heidkamp,[1] Melinda Kay Munos[1]

[1]Department of International Health, Johns Hopkins University Bloomberg School of Public Health, Baltimore, Maryland, USA
[2]Department of Biostatistics, Johns Hopkins University Bloomberg School of Public Health, Baltimore, Maryland, USA

**Correspondence to**
Dr Ashley Sheffel, Department of International Health, Johns Hopkins Bloomberg School of Public Health, Baltimore, MD, USA; asheffel@jhu.edu

## ABSTRACT

**Introduction** Measuring quality of care in low-income and middle-income countries is complicated by the lack of a standard, universally accepted definition for 'quality' for any particular service, as well as limited guidance on which indicators to include in measures of quality of care, and how to incorporate those indicators into summary indices. The aim of this paper is to develop, characterise and compare a set of antenatal care (ANC) indices for facility readiness and provision of care.

**Methods** We created nine indices for facility readiness using three methods for selecting items and three methods for combining items. In addition, we created three indices for provision of care using one method for selecting items and three methods for combining items. For each index, we calculated descriptive statistics, categorised the continuous index scores using tercile cut points to assess comparability of facility classification, and examined the variability and distribution of scores.

**Results** Our results showed that, within a country, the indices were quite similar in terms of mean index score, facility classification, coefficient of variation, floor and ceiling effects, and the inclusion of items in an index with a range of variability. Notably, the indices created using principal components analysis to combine the items were the most different from the other indices. In addition, the index created by taking a weighted average of a core set of items had lower agreement with the other indices when looking at facility classification.

**Conclusions** As improving quality of care becomes integral to global efforts to produce better health outcomes, demand for guidance on creating standardised measures of service quality will grow. This study provides health systems researchers with a comparison of methodologies commonly used to create summary indices of ANC service quality and it highlights the similarities and differences between methods.

## INTRODUCTION

Reducing maternal morbidity and mortality has been a global priority over the last several decades. While progress has been made in both improving coverage and reducing mortality, there remains an unacceptably high number of deaths of women globally.[1] In many countries, considerable increases in the coverage of health services have not translated into sufficient reductions in mortality.[2]

### Strengths and limitations of this study

► To our knowledge, this is the first study that uses multiple methodologies to develop and compare indices for antenatal care (ANC) facility readiness and provision of care.
► Our approaches to item selection and combination were based on the most commonly used approaches in the literature and we included those most likely to be useful to low-income and middle-income countries.
► The analysis was conducted using data from multiple countries, which strengthens our ability to generalise the findings to other similar contexts.
► The Service Provision Assessment survey, while quite comprehensive, does not capture every aspect of quality of care that may be important to include in quality of care metrics.
► This analysis does not provide a formal validation of any one ANC quality of care index.

This finding suggests that quality of care may play a critical role in producing better health outcomes. Reducing maternal mortality will require increasing the coverage of health interventions, and ensuring that those interventions are delivered with a high level of quality.[3 4] These interventions include intrapartum and postpartum care, and antenatal care (ANC), which is important to maintaining a healthy pregnancy,[5–7] and promotes safe delivery, postnatal attendance, and is positively associated with an increase in facility-based deliveries.[8–10]

Low-income and middle-income countries (LMICs) are increasingly shifting focus towards making improvements in the quality of health services delivered, which requires measuring quality of care and monitoring progress. However, for LMICs to systematically and comprehensively measure quality of care, they require a definition of quality of care and clear, specific indicators to operationalise the definition. Many quality of care frameworks exist, most of which build from Donabedian's 1988 framework characterising

quality at three levels: structure, commonly called facility readiness (the setting in which care occurs including material resources, human resources and organisational structure), process (commonly called provision of care—the quality of medical advice delivered by providers to clients, as well as interpersonal relationships between the provider and the client), and outcome (the effects of care on the health status and behaviours of patients, as well as improvements in patient knowledge and the degree of satisfaction with care).[11 12] While there may be growing consensus on the core components of a framework for describing quality of care, measuring quality of care is complicated by lack of a standard, universally accepted definition for 'quality' for any one particular service.[13–15] In addition, there is limited guidance on which indicators to include in measures of quality of care and how to incorporate those indicators into summary indices.

Studies on quality of care in LMICs largely rely on data from health facility surveys that collect data on facility readiness and provision of care. These studies often use summary indices or composite scores to provide an overall description of service quality.[16–26] Summary scores are useful to LMIC governments in that they simplify complex data and enable comparison of performance within facilities, across administrative units and over time. However, there is little consistency between studies in terms of the items included in summary metrics of quality of care, or in how the metrics are created. A scoping review published in 2018 identified numerous studies that used health facility survey data to assess maternal and newborn quality of care in LMICs and found that studies used various approaches to create quality of care metrics.[27] Online supplementary table 1 illustrates this with an excerpt of studies that used data from the Service Provision Assessment (SPA) to create quality of care metrics for ANC specifically, each using a different approach. Item selection was guided by different sources including the Donabedian quality of care framework,[17 28] clinical guidelines[18 19 29 30] and various definitions of quality of care[20 21]. The number of items that were included in an index varied from a minimum of 7 to a maximum of 40, and various methods were used to combine items into indices. These methods include a simple and weighted additive approaches[20 22–26] as well as principal components analysis (PCA)[28 31–33]. While many of these studies had some overlap in the items selected for inclusion in their ANC

quality of care index, the quality of care indices varied greatly in the items selected for inclusion in the indices and in the methodology used for combining items into a summary measure.

As quality of care indices are increasingly used by countries, global agencies and researchers to quantify the quality of health services and estimate effective coverage, there is a need to standardise the methodology for creating these measures. To our knowledge, no studies have compared the methodologies used to create indices of ANC service quality. The objective of this paper is to develop, characterise and compare a set of facility readiness and provision of care indices for ANC.

## METHODS
### Overall approach
Using data from an expert survey, we created nine facility readiness indices using three methods for selecting items (a 'core set' of items, 'expert survey' set of items and 'maximum' set of items) and three methods for combining items (simple additive, weighted additive and PCA) (figure 1). We created three indices for provision of care using one method for selecting items ('expert survey' set of items) and three methods for combining items (simple additive, weighted additive and PCA) (figure 1). Data for these indices come from the SPA. We then compared the indices and examined the variability and distribution of the index scores.

### Data source
The SPA is a health facility assessment used in LMICs to generate nationally representative data on health service delivery.[34] While there are a number of widely used health facility assessment tools, we chose to use SPA data for this analysis as the SPA has been conducted in a number of different countries, includes observation of clinical care, and the data are publicly available through the Demographic and Health Survey (DHS) programme. The SPA includes a standard set of survey instruments: a facility inventory questionnaire, health worker interviews, observation of ANC consultations and exit interviews with ANC clients.

We examined all SPA surveys for inclusion in the analysis (total of 16). We included all SPA surveys which used the DHS-VI or DHS-VII questionnaire (four surveys

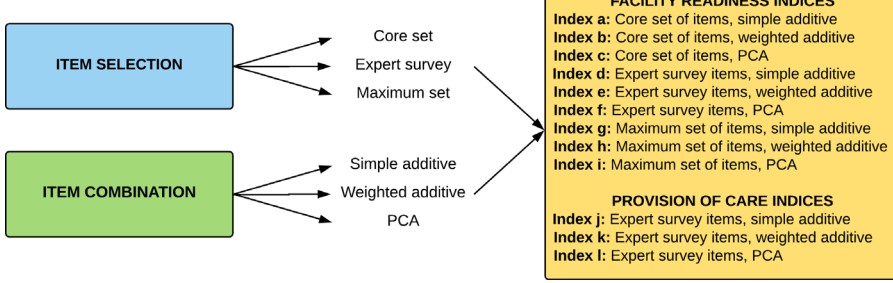

**Figure 1** Process for index creation. PCA, principal components analysis.

excluded), that included observations of ANC consultations (three surveys excluded), were conducted in the last 5 years (between 2013 and 2018) (two surveys excluded), and for which recode data files were available (four surveys excluded). The included surveys are from Haiti (2013), Malawi (2013/2014) and Tanzania (2014/2015). The SPA final country reports contain comprehensive information on the survey methodology and questionnaires.[35–37] Briefly described, the survey in Tanzania was a nationally representative sample of health facilities. Health facilities were selected using stratified systematic probability sampling with stratification by region and facility type (with oversampling of some facility types such as hospitals). In Haiti and Malawi, the survey was a census of all health facilities in the country. In all three countries, all surveyed facilities completed the facility inventory module. In addition, within each facility, up to eight health workers were interviewed including all health workers whose consultations were observed and those who provided information for any section of the facility questionnaire. ANC clients were selected for observation using systematic sampling, based on the number of clients present at each service site on the day of the visit. At facilities where the number of ANC clients expected on the day of the survey could not be predetermined, the sample was opportunistic since clients were selected as they arrived. Observation was completed for a minimum of five clients per service provider, with a maximum of 15 observations in any given facility for each service. Client exit interviews were conducted with every client whose visit was observed.

## Item selection

To identify elements of facility readiness and the provision of ANC considered by experts to be the most important to ANC service quality, we reviewed the Service Availability and Readiness Assessment (SARA) indicators, World Health Organization (WHO) Focused Antenatal Care (FANC) guidelines and WHO recommendations on ANC for a positive pregnancy experience, as well as the SPA questionnaire. This process identified a total of 121 items organised according to the dimensions of quality of care proposed by the WHO Quality of Care Framework for maternal and newborn health: essential physical resources (41 items), competent and motivated human resources (four items), provision of care (65 items) and experience of care (11 items).[38]

We then conducted a survey of 50 maternal health experts who had experience working in LMICs (questionnaire available as an online supplementary file). Respondents were asked to rate each item based on overall importance. Importance ratings ranged from one (item was unimportant) to four (item was essential). Experts also provided a list of items they felt were important for delivering high-quality ANC services, but were missing from the survey instrument. Fifteen maternal health experts completed the survey, with respondents representing academic institutions, donor agencies, United Nations agencies, global health implementing organisations and Ministries of Health in LMICs.

Items that were not collected across SPA surveys in all three countries and items that are not required for a first ANC visit were excluded from all indices. For each readiness item, a binary score was created based on whether the criteria for availability were met on the day of the survey (1=available, 0=not available). For human resources items, each item is a proportion ranging from 0 to 1. For provision of care items, a binary score was created based on whether or not the activity was conducted during the ANC visit (1=conducted, 0=not conducted) or whether or not the client had a problem with the item during the ANC visit (1=no problem, 0=major or minor problem). The list of items included in each readiness and provision of care index can be found in online supplementary tables 2 and 3.

Three methods were used to select items to include in the facility readiness indices. The first method identified the core set of readiness items required to deliver ANC services. This set of items was identified by reviewing the provision of care items required for an ANC visit based on WHO FANC guidelines and WHO recommendations on ANC for a positive pregnancy experience, and by determining the human resources, equipment and supplies, medicines, and diagnostics required to deliver each specific item. In creating the core set of items for facility readiness, we mapped each provision of care item to the facility readiness items required to deliver the specific service component. We found that of the 49 provision of care items, 36 items required only human resources and 13 items required human resources plus equipment, diagnostics, medicines or basic amenities. The core index did not include standard precautions for infection prevention items because these are not explicitly required for any one provision of care item. A total of 21 items were selected for the core set readiness index.

The second method used results from the expert survey to identify the set of readiness items maternal health experts identified as essential to deliver ANC services. Mean ratings were calculated for each readiness item (see online supplementary table 4 for results). Items rated by the expert group with a mean importance of >3.4 out of 4 were selected for the expert survey index. The threshold for inclusion was determined by examining the distribution of scores and identifying a natural break point which separated the top-rated items from the rest. In addition, items were selected so that at least one item per domain (human resources, equipment and supplies, diagnostics, medicines, basic amenities) was included in the index. A total of 19 items were selected, representing 42% of the total items for the expert survey readiness index.

The third method identified the maximum set of readiness items used to deliver ANC services. This index included all items identified in the SPA related to ANC readiness across the following domains: human resources, equipment and supplies, medicines, diagnostics and basic amenities. Out of the 45 facility readiness items identified

**Table 1** Descriptive statistics of Facility Readiness and Provision of Care Indices

**Facility Readiness Indices**

| | Core simple | Core weighted | Core PCA | Expert simple | Expert weighted | Expert PCA | Maximum simple | Maximum weighted | Maximum PCA |
|---|---|---|---|---|---|---|---|---|---|
| **Haiti (n=358 facilities)** | | | | | | | | | |
| Mean | 63.8 | 64.0 | 59.4 | 64.1 | 61.3 | 58.2 | 57.8 | 56.6 | 59.8 |
| SD | 14.0 | 15.1 | 21.4 | 16.1 | 16.6 | 21.6 | 12.6 | 14.0 | 15.9 |
| Median | 63.3 | 65.5 | 59.4 | 64.3 | 61.3 | 57.5 | 57.9 | 57.4 | 60.6 |
| Minimum | 23.8 | 16.4 | 17.6 | 17.1 | 18.1 | 12.9 | 27.6 | 20.1 | 23.4 |
| Maximum | 92.9 | 93.8 | 99.8 | 100.0 | 100.0 | 100.0 | 87.5 | 87.7 | 95.2 |
| IQR | 19.8 | 20.7 | 40.9 | 24.6 | 24.2 | 39.5 | 17.8 | 20.2 | 24.6 |
| **Malawi (n=253 facilities)** | | | | | | | | | |
| Mean | 62.9 | 65.2 | 52.3 | 68.7 | 67.6 | 69.6 | 61.3 | 61.0 | 63.1 |
| SD | 13.2 | 13.3 | 16.7 | 13.2 | 13.1 | 13.2 | 11.5 | 11.4 | 12.9 |
| Median | 61.9 | 66.4 | 46.4 | 68.4 | 67.8 | 66.3 | 59.9 | 60.0 | 59.8 |
| Minimum | 33.3 | 30.0 | 24.5 | 38.2 | 34.3 | 38.9 | 35.5 | 32.7 | 37.0 |
| Maximum | 100.0 | 100.0 | 100.0 | 100.0 | 100.0 | 100.0 | 92.8 | 94.3 | 100.0 |
| IQR | 19.0 | 16.7 | 20.5 | 17.5 | 16.9 | 16.5 | 16.7 | 14.1 | 18.7 |
| **Tanzania (n=632 facilities)** | | | | | | | | | |
| Mean | 71.4 | 71.0 | 70.2 | 75.9 | 71.8 | 74.7 | 61.2 | 62.8 | 64.1 |
| SD | 13.6 | 14.6 | 15.5 | 15.4 | 14.6 | 17.8 | 13.0 | 13.6 | 14.6 |
| Median | 73.5 | 72.1 | 73.9 | 78.9 | 73.2 | 78.4 | 61.8 | 63.9 | 64.9 |
| Minimum | 23.8 | 18.6 | 20.4 | 28.9 | 28.6 | 20.3 | 27.6 | 25.5 | 24.7 |
| Maximum | 100.0 | 100.0 | 100.0 | 100.0 | 100.0 | 100.0 | 94.7 | 92.1 | 100.0 |
| IQR | 21.4 | 20.3 | 25.8 | 23.8 | 20.2 | 30.2 | 20.2 | 19.9 | 23.0 |

**Provision of Care Indices**

| | Simple | Weighted | PCA |
|---|---|---|---|
| **Haiti (n=779 clients)** | | | |
| Mean | 38.9 | 45.0 | 85.9 |
| SD | 8.0 | 10.4 | 15.9 |
| Median | 38.8 | 45.4 | 91.7 |
| Minimum | 12.2 | 14.0 | 0.0 |
| Maximum | 79.6 | 76.3 | 96.0 |
| IQR | 8.5 | 12.9 | 7.1 |
| **Malawi (n=815 clients)** | | | |
| Mean | 47.5 | 53.1 | 35.8 |
| SD | 9.9 | 10.2 | 14.5 |
| Median | 46.9 | 53.7 | 33.9 |
| Minimum | 22.4 | 22.9 | 10.7 |
| Maximum | 72.0 | 75.5 | 70.8 |
| IQR | 14.3 | 13.5 | 24.5 |
| **Tanzania (n=1681 clients)** | | | |
| Mean | 50.8 | 58.0 | 40.4 |
| SD | 10.9 | 11.2 | 12.3 |
| Median | 50.5 | 58.6 | 39.5 |
| Minimum | 16.3 | 18.5 | 9.9 |
| Maximum | 98.0 | 99.3 | 98.2 |
| IQR | 14.8 | 15.1 | 17.0 |

IQR, interquartile range; PCA, principal components analysis; SD, standard deviation.

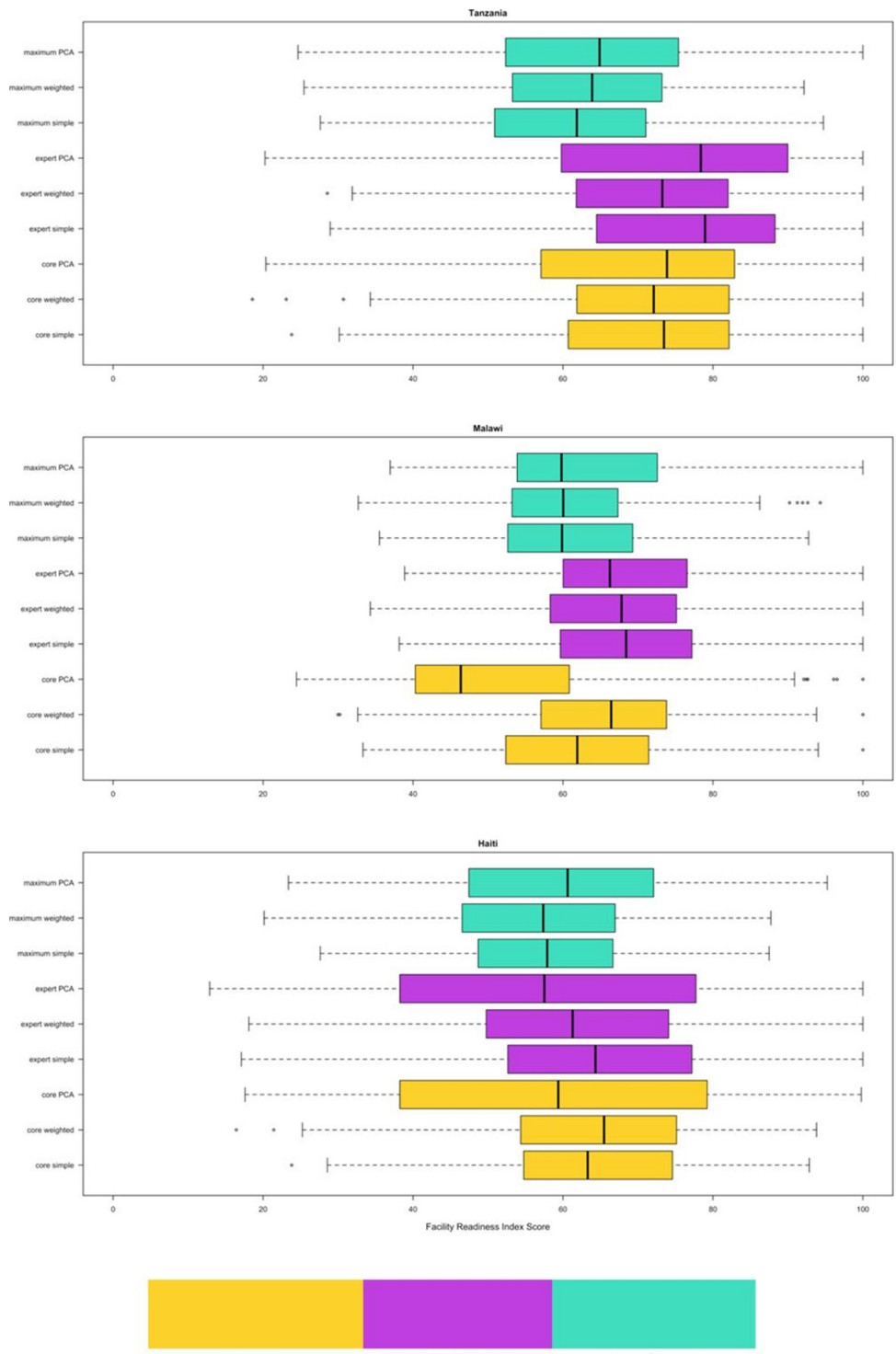

**Figure 2** Distribution of facility readiness scores. PCA, principal components analysis.

for inclusion in the expert survey, a total of 38 items were selected for the maximum set readiness index. Seven items from the expert survey were not included in the maximum set of readiness items as data was not collected in the SPA on these items.

We used a single method to select items to include in the provision of care index. The method used results from the <u>expert survey</u> to select the set of provision of care items maternal health experts identified as essential

to deliver ANC services. We chose this method because maternal health experts were the best source for determining which processes are essential to high-quality ANC consultations. In addition, the experts selected most items as very important or essential, and therefore, it was not appropriate to define a core and maximum set of items. Mean ratings were calculated for each provision of care item (see online supplementary table 4 for results). Items rated by the expert group with a mean importance

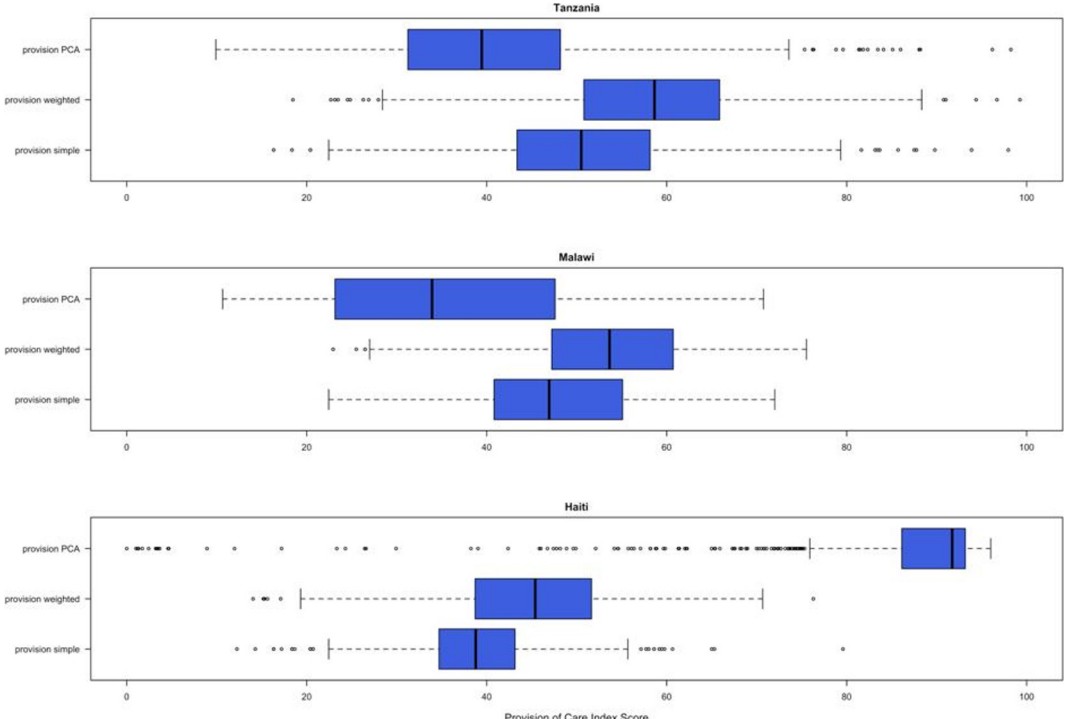

**Figure 3** Distribution of provision of care scores. PCA, principal components analysis.

of ≥3.0 out of 4 were selected for the provision of care index. The threshold for inclusion was decided by examining the distribution of scores which showed a bimodal distribution; then, selecting all items in the first mode. The expert survey respondents mentioned a number of provision of care items, largely related to experience of care, that they felt were essential for high-quality care, but were missing from our survey and from SPA datasets. A total of 49 items were selected representing 64% of the total items for the provision of care index.

### Item combination
Three methods were used to combine items to create the facility readiness and provision of care indices—simple additive, weighted additive and PCA.

### Simple additive
The simple additive index score was calculated by taking a sum of the items available divided by the total number of items in the index. We transformed the index into a score out of 100 by dividing by the number of items and multiplying by 100. The simple additive index weighted all items in the index equally.

### Weighted additive
The weighted additive index was also calculated as a sum of items, but instead of assuming equal weights for all items, the weighted additive index accounted for the number of items within each domain. Readiness items were first grouped into five domains: human resources, equipment and supplies, medicines, diagnostics, and basic amenities. For the provision of care index, items were initially grouped into five domains: history and

counselling, examination, diagnostics, preventative treatment and client experience. We then computed a domain score by adding the items within each domain and dividing by the total number of items in the domain. Finally, we transformed the index into a score out of 100 by averaging the domain scores and multiplying by 100.

### Principal components analysis
To create the PCA index score, we conducted an unrotated, unweighted PCA using a correlation matrix and used the factor loadings from the first principal component to create the index score. We rescaled the score obtained from the first component of the PCA to a range of 0–100 for comparability with other indices.

### Analysis
We limited the readiness analysis to facilities offering ANC services and with at least one first visit ANC client observation. In order to standardise expected clinical actions, we limited the provision of care analysis to women attending the health facility for their first ANC visit. Finally, we excluded cases with incomplete data for any variables of interest.

We calculated descriptive statistics on each of the indices including mean, median, minimum, maximum and range. The elements of the complex survey design (weights and clustering of observations) were not incorporated into the analysis as the goal of this analysis was not to make inferences about the entire population from which the sample of health facilities was drawn. In addition, for the indices created using PCA to combine items, we calculated factor loadings, the eigenvalue and the per

cent of total variance explained by the first component. Since the PCA method was used to create weights for a composite variable rather than estimating a latent variable we did not report out the number of components with an eigenvalue greater than one. Next, to compare scores from the nine facility readiness indices, we categorised the continuous index scores into low, medium and high readiness categories using tercile cut points. We then assessed the comparability of the facility classification across the indices by calculating the per cent agreement and Cohen's kappa, which accounts for the possibility of the agreement occurring by chance, between each combination of indices. Cohen's kappa ranges from −1 to 1 and can be interpreted as follows: <0 as no agreement, 0–0.20 as slight, 0.21–0.40 as fair, 0.41–0.60 as moderate, 0.61–0.80 as good and 0.81–1 as almost perfect agreement.[39]

Next, we examined the variability and distribution of the index scores. As countries are interested in being able to compare facilities with each other to understand better and worse performers (even if they are all within the low-quality or high-quality band), we are interested in being able to capture that variation in an index score. The examination of variability and distribution of the index scores aims to understand the level of variability being captured but does not indicate that an index is necessarily 'better' because it captures more variability. The coefficient of variation (CoV) was used to measure the variation captured by each index, with a higher CoV indicative of more variation captured in the index. Distribution of the scores was examined by assessment of floor and ceiling effects. Floor and ceiling effects were considered to be present if more than 15% of facilities achieved the lowest or highest possible index score (0% or 100%), respectively.[40] While in some settings it may be possible that greater than 15% of facilities are at 0% or 100%, the presence of floor and ceiling effects may indicate that indices have limited ability to differentiate facilities at very low and very high readiness levels. Finally, each index was assessed in terms of inclusion of items with a range of variability. For each index, we tallied the number of items that were available in <30% of facilities, <40% of facilities and >90% of facilities.

All statistical analyses were carried out using R V.3.5.1.[41]

### Patient and public involvement
It was not appropriate or possible to involve patients or the public in this work.

### RESULT
The final analytical sample consisted of 358 health facilities and 779 ANC clients in Haiti, 253 health facilities and 815 ANC clients in Malawi, and 632 health facilities and 1681 ANC clients in Tanzania. A total of 6 ANC clients in Haiti, 44 ANC clients in Malawi and 73 ANC clients in Tanzania were excluded from the analysis due to missing data.

### Item selection
The results of the expert survey are available in online supplementary table 4. There was some alignment between the core set of readiness items and the expert set of readiness items. The core readiness index included 21 items, the expert readiness index included 19 items and the overlap between the two indices was 14 items. Of the 45 facility readiness items identified for inclusion in the expert survey, only four were related to human resources and of those, only one was highly rated by the expert group as essential for high-quality service delivery. In addition, the expert survey respondents mentioned a number of items that they felt were essential for high-quality care but were missing from our survey and from SPA datasets. These items included: ensuring that clients are treated with respect and without discrimination, having the option for clients to invite their partner to participate in the consultation, discussing identification of a birth companion of choice with the client, including family members in counselling sessions, providing counselling on newborn care practices and ensuring clients understand the counselling information they receive.

### Descriptive statistics
The mean readiness score varied by index from 56.6 to 64.1 in Haiti, 52.3 to 69.6 in Malawi and 61.2 to 75.9 in Tanzania (table 1, figure 2). Across all countries, there were no indices for which any facilities received the minimum score of 0 and all countries had at least three indices for which some facilities received the maximum score of 100. In addition, in each country, the three indices that contained the same items—but used different methodologies for combining the items—generally had a similar median. For example, the core simple, core weighted and core PCA median scores in Tanzania were 73.5, 72.1 and 73.9, respectively. Finally, in general, the indices created using PCA resulted in a larger IQR, particularly for the core and expert item indices, across all three countries.

The mean provision of care score varied by index from 38.9 to 85.9 in Haiti, 35.8 to 53.1 in Malawi and 40.4 to 58.0 in Tanzania (table 1, figure 3). The expert PCA index in Haiti was the only index in which any clients received the minimum score of 0 and no countries had an index for which any clients received the maximum score of 100. The expert simple and expert weighted indices had similar median provision of care scores; however, the expert weighted index scores were higher than the simple index scores in all three countries. The expert PCA index scores had median provision of care scores which were very different from both the expert simple and expert weighted scores. For example, in Malawi, the expert simple median provision of care score was 46.9, the expert weighted median provision of care score was 53.7 and the expert PCA median provision of care score was 33.9.

The results from the PCA analysis are presented in table 2. Across all countries, and all indices for facility readiness, the items with the highest loadings (absolute

**Table 2** Results from the Facility Readiness Indices and Provision of Care Indices principal component analyses

| Item name | Haiti | | | Malawi | | | Tanzania | | |
|---|---|---|---|---|---|---|---|---|---|
| | Core set | Expert set | Maximum set | Core set | Expert set | Maximum set | Core set | Expert set | Maximum set |
| **Facility readiness indices** | | | | | | | | | |
| **Equipment and supplies** | | | | | | | | | |
| Blood pressure apparatus | 0.05 | 0.04 | 0.04 | 0.37 | 0.34 | 0.37 | 0.34 | 0.33 | 0.28 |
| Examination light | 0.08 | | 0.14 | 0.44 | | 0.49 | 0.34 | | 0.38 |
| Stethoscope | 0.14 | 0.11 | 0.17 | 0.16 | 0.13 | 0.18 | 0.28 | 0.23 | 0.23 |
| Adult weighing scale | 0.22 | | 0.19 | 0.25 | | 0.32 | 0.43 | | 0.39 |
| Tape measure for fundal height | 0.13 | | 0.18 | 0.22 | | 0.28 | 0.21 | | 0.19 |
| Examination bed | 0.23 | 0.22 | 0.26 | 0.17 | 0.21 | 0.18 | 0.14 | 0.10 | 0.15 |
| Latex gloves | 0.24 | 0.52 | 0.49 | 0.14 | 0.32 | 0.39 | 0.08 | 0.50 | 0.51 |
| Single use syringes | | 0.25 | 0.26 | | 0.24 | 0.21 | | 0.14 | 0.14 |
| Soap and running water or alcohol-based hand rub | | 0.41 | 0.41 | | 0.38 | 0.40 | | 0.52 | 0.57 |
| Disinfectant | | 0.40 | 0.43 | | 0.34 | 0.30 | | 0.32 | 0.30 |
| Appropriate storage of sharps waste (sharps box) | | | −0.12 | | | 0.09 | | | 0.11 |
| Appropriate storage of infectious waste (pedal bin with lid) | | | 0.01 | | | 0.03 | | | 0.17 |
| Safe final disposal of sharps (incineration) | | | 0.20 | | | 0.26 | | | 0.06 |
| Safe final disposal of infectious wastes (incineration) | | | 0.20 | | | 0.25 | | | 0.06 |
| Medical masks | | | 0.06 | | | 0.16 | | | 0.35 |
| Gowns | | | 0.31 | | | 0.21 | | | 0.38 |
| Eye protection | | | 0.29 | | | 0.28 | | | 0.23 |
| **Diagnostics** | | | | | | | | | |
| Haemoglobin | 0.63 | 0.61 | 0.60 | 0.75 | 0.75 | 0.70 | 0.72 | 0.71 | 0.69 |
| Urine dipstick-protein | 0.78 | 0.77 | 0.69 | 0.85 | 0.84 | 0.77 | 0.77 | 0.78 | 0.68 |
| Urine dipstick-glucose | 0.74 | 0.73 | 0.66 | 0.80 | 0.78 | 0.71 | 0.78 | 0.79 | 0.69 |
| Grouping and rhesus factor | 0.18 | | 0.19 | 0.53 | | 0.47 | 0.29 | | 0.37 |
| Syphilis RDT/RPR | 0.78 | 0.77 | −0.04 | 0.68 | 0.68 | −0.05 | 0.65 | 0.65 | −0.04 |
| HIV testing/RDT | 0.78 | 0.79 | 0.72 | 0.18 | 0.23 | 0.21 | 0.41 | 0.41 | 0.35 |
| **Medicines** | | | | | | | | | |
| Iron tablets | 0.10 | 0.11 | 0.18 | 0.11 | 0.17 | 0.15 | 0.26 | 0.26 | 0.25 |
| Folic acid tablets | 0.15 | 0.16 | 0.19 | 0.16 | 0.24 | 0.22 | 0.20 | 0.21 | 0.21 |
| TT vaccine | −0.05 | −0.02 | 0.01 | 0.20 | 0.25 | 0.24 | 0.33 | 0.30 | 0.27 |
| Deworming drugs | 0.09 | | 0.12 | −0.12 | | −0.06 | 0.15 | | 0.09 |
| **Basic amenities** | | | | | | | | | |
| Power | | 0.18 | 0.55 | | 0.17 | 0.29 | | 0.44 | 0.41 |
| Improved water source | | −0.08 | 0.23 | | 0.10 | 0.14 | | 0.03 | 0.39 |

Continued

**Table 2** Continued

| Item name | Haiti | | | Malawi | | | Tanzania | | |
|---|---|---|---|---|---|---|---|---|---|
| | Core set | Expert set | Maximum set | Core set | Expert set | Maximum set | Core set | Expert set | Maximum set |
| Room with auditory and visual privacy | −0.10 | 0.46 | −0.07 | 0.06 | 0.20 | 0.06 | 0.08 | 0.26 | 0.06 |
| Sanitation facilities | | | 0.42 | | | 0.10 | | | 0.25 |
| Communication equipment | | | 0.25 | | | 0.27 | | | 0.60 |
| Computer with email/internet | | | 0.62 | | | 0.71 | | | 0.66 |
| Emergency transportation | | | 0.38 | | | 0.01 | | | 0.33 |
| **Human resources** | | | | | | | | | |
| Guidelines ANC | −0.06 | | 0.00 | 0.15 | | 0.20 | 0.38 | 0.33 | 0.36 |
| Proportion of staff trained in ANC in last 2 years | 0.43 | 0.39 | 0.39 | 0.47 | 0.46 | 0.44 | 0.40 | | 0.34 |
| Proportion of staff receiving supervision in the last 6 months | 0.60 | | 0.59 | 0.43 | | 0.40 | 0.42 | | 0.37 |
| Proportion of staff reporting opportunities for promotion | | | 0.17 | | | −0.05 | | | −0.07 |
| **Eigenvalue** | 3.61 | 3.87 | 3.61 | 3.74 | 3.42 | 3.74 | 3.66 | 3.70 | 3.66 |
| **Per cent of total variance** | 17.21 | 20.38 | 17.21 | 17.80 | 17.98 | 17.80 | 17.44 | 19.49 | 17.44 |

RDT, Rapid Diagnostic Test; RPR, Rapid Plasma Reagin; TT, tetanus toxoid.

**Provision of Care Indices**

| Item name | Haiti | Malawi | Tanzania |
|---|---|---|---|
| **History and counselling** | | | |
| History taking: personal history: client age | 0.03 | 0.24 | 0.28 |
| History taking: personal history: medications client is taking | −0.02 | 0.05 | 0.22 |
| History taking: personal history: date last menstrual period began | 0.02 | 0.25 | 0.32 |
| History taking: personal history: any prior pregnancy | −0.09 | 0.44 | 0.31 |
| History taking: medical history for prior pregnancies: stillbirth | −0.18 | 0.69 | 0.46 |
| History taking: medical history for prior pregnancies: infant died in the first week of life | −0.01 | 0.59 | 0.45 |
| History taking: medical history for prior pregnancies: heavy bleeding during or after delivery | 0.03 | 0.73 | 0.50 |
| History taking: medical history for prior pregnancies: previous assisted delivery | −0.02 | 0.73 | 0.42 |
| History taking: medical history for prior pregnancies: previous spontaneous abortion | 0.00 | 0.69 | 0.39 |
| History taking: medical history for prior pregnancies: previous multiple pregnancies | 0.02 | 0.54 | 0.37 |
| History taking: medical history for prior pregnancies: previous prolonged labour | 0.00 | 0.44 | 0.39 |
| History taking: medical history for prior pregnancies: pregnancy-induced hypertension | −0.13 | 0.73 | 0.35 |
| History taking: medical history for prior pregnancies: pregnancy-related convulsions | 0.00 | 0.68 | 0.37 |
| History taking: medical history for prior pregnancies: high fever or infection during prior pregnancy/pregnancies | −0.04 | 0.21 | 0.35 |

**Table 2** Continued

**Provision of Care Indices**

| Item name | Haiti | Malawi | Tanzania |
|---|---|---|---|
| History taking: history of complaints in current pregnancy: vaginal bleeding | 0.01 | 0.31 | 0.58 |
| History taking: history of complaints in current pregnancy: fever | 0.07 | 0.15 | 0.51 |
| History taking: history of complaints in current pregnancy: headache or blurred vision | 0.03 | 0.18 | 0.56 |
| History taking: history of complaints in current pregnancy: swollen face or hands or extremities | 0.03 | 0.25 | 0.48 |
| History taking: history of complaints in current pregnancy: tiredness or breathlessness | −0.02 | 0.17 | 0.51 |
| History taking: history of complaints in current pregnancy: fetal movement (loss of, excessive, normal) | 0.04 | 0.10 | 0.47 |
| History taking: history of complaints in current pregnancy: cough or difficulty breathing for 3 weeks or longer | −0.04 | 0.21 | 0.36 |
| Client education and counselling: process of pregnancy and its complications | 0.01 | 0.16 | 0.27 |
| Client education and counselling: diet and nutrition | 0.03 | 0.20 | 0.28 |
| Client education and counselling: danger signs in pregnancy | −0.01 | 0.14 | 0.65 |
| Client education and counselling: voluntary counselling and testing for HIV | 0.08 | 0.12 | 0.27 |
| Client education and counselling: breast feeding | −0.02 | 0.07 | 0.29 |
| Client education and counselling: plans of delivery (emergency preparedness, place of delivery, transportation, financial arrangements) | 0.07 | 0.28 | 0.37 |
| **Examination** | | | |
| Examination and observation: oedema (other than ankle specify) | 0.08 | 0.21 | 0.24 |
| Examination and observation: blood pressure | 0.03 | 0.09 | 0.10 |
| Examination and observation: weight | 0.06 | 0.11 | 0.16 |
| Obstetric complications: palpate the client's abdomen for fundal height | 0.15 | 0.11 | 0.19 |
| **Diagnostics** | | | |
| Laboratory investigations: haemoglobin | 0.03 | −0.01 | 0.12 |
| Laboratory investigations: grouping and rhesus factor | 0.04 | 0.15 | 0.16 |
| Laboratory investigations: RPR (syphilis test) | 0.03 | 0.04 | 0.16 |
| Laboratory investigations: HIV testing | 0.04 | 0.04 | 0.23 |
| Laboratory investigations: urine protein, sugar, acetone | 0.03 | 0.03 | 0.09 |
| **Preventative treatment** | | | |
| Drug administration and immunisation: Iron and/or folic acid provided or prescribed | 0.06 | 0.13 | 0.18 |
| Drug administration and immunisation: TT provided or prescribed | 0.06 | 0.03 | 0.09 |
| **Client experience** | | | |
| No problem: discuss problems or concerns | 0.65 | −0.01 | 0.14 |
| No problem: explanation you received | 0.64 | 0.08 | 0.15 |
| No problem: how the staff treated you | 0.73 | 0.05 | 0.09 |
| No problem: privacy from having others see | 0.72 | 0.07 | 0.20 |
| No problem: privacy from having others hear | 0.74 | 0.10 | 0.20 |
| No problem: time you waited | 0.44 | 0.04 | 0.01 |
| No problem: number of days services are available | 0.73 | 0.01 | 0.11 |
| No problem: hours of service | 0.75 | 0.00 | 0.06 |
| No problem: cost for services or treatments | 0.65 | 0.10 | 0.08 |
| No problem: availability of medicines | 0.62 | 0.02 | 0.06 |

Continued

**Table 2** Continued

**Provision of Care Indices**

| Item name | Haiti | Malawi | Tanzania |
|---|---|---|---|
| No problem: cleanliness of the facility | 0.69 | 0.03 | 0.06 |
| Eigenvalue | 5.14 | 4.84 | 5.08 |
| Per cent of total variance | 10.49 | 9.89 | 10.37 |

ANC, antenatal care; RDT, Rapid Diagnostic Test; RPR, Rapid Plasma Reagin; TT, tetanus toxoid.

value greater than 0.4) differed by index and country. For the core item indices, the items that loaded the highest were related to diagnostics and human resources. For the expert item indices, the items that loaded the highest were related to diagnostics. For the maximum item indices, the items that loaded the highest were related to diagnostics and basic amenities. For all facility readiness indices in all countries, the per cent of variance explained by the first principal component was low, ranging from 17.21% for the core and maximum indices in Haiti to 19.49% and 20.38% for the expert set of items in Tanzania and Haiti, respectively. In addition, we found that some items, such as tetanus toxoid vaccine, deworming medications and syphilis testing, had negative loadings.

Across all countries for provision of care, the items with the highest loadings (absolute value greater than 0.4) differed by country. For Haiti, the items that loaded the highest were related to client experience. For Malawi and Tanzania, the items that loaded the highest were related to history taking and counselling; although for each country, the highest loadings were largely on different items. For provision of care indices in all countries, the per cent of variance explained by the first principal component was low, ranging from 9.89% in Malawi to 10.49% in Haiti. In addition, we found that items such as history taking, client education and counselling on danger signs in pregnancy, and haemoglobin testing had negative loadings.

### Comparability of the facility classification

Table 3 presents the results of the per cent agreement and Cohen's kappa coefficient among the nine facility readiness indices. Across all countries, all readiness indices had fair or better agreement. In Haiti, there were three index combinations that had fair agreement (core weighted/core PCA; core weighted/expert PCA; core weighted/maximum PCA) while the remainder of the indices had moderate or better agreement. In Malawi, there was one index combination that had fair agreement (core weighted/maximum simple) while the remainder of the indices had moderate or better agreement. In Tanzania, there were two index combinations that had fair agreement (core weighted/expert PCA; core weighted/maximum PCA) while the remainder of the indices had moderate or better agreement.

### Variability and distribution of the index

The variability and distribution of the nine facility readiness indices are presented in table 4. In Haiti, the CoV ranged from 21.12 (maximum simple readiness index) to 35.72 (expert PCA). In Malawi, the CoV ranged from 18.96 (expert PCA) to 32.09 (core PCA). In Tanzania, the CoV ranged from 19.11 (core simple) to 23.62 (expert PCA). Across all countries and all indices, there were no floor effects. Ceiling effects were limited and far below the 15% threshold. The highest percentage of ceiling effects was 3.33%, which was found in Tanzania in three indices (expert simple, expert weighted and expert PCA).

### Inclusion of items across a range of frequency

Table 5 presents the percentage of facilities in which each index item was available in order to assess the inclusion of items across a range of frequency. Across all countries, the maximum item index contained the greatest number of items available in less than 40% of facilities (10 items Haiti, 12 items Malawi and 9 items in Tanzania). In Tanzania, the expert index did not include any items available in less than 40% of facilities. Within countries, the core, expert and maximum indices had a similar number of items that were available in over 90% of facilities (four to five in Haiti, six to eight in Malawi and seven to nine in Tanzania).

### DISCUSSION

To our knowledge, this is the first study that uses multiple methodologies to develop and compare indices for ANC facility readiness and provision of care. Our results showed that, within a country, indices containing different items combined using different approaches were quite similar in terms of median index score, facility classification, CoV, floor and ceiling effects and inclusion of items with a range of variability. Although similar overall, we found that the indices created using PCA were the most different from the other indices. In addition, the core weighted index had lower agreement with the other indices when looking at facility classification. This may be the result of some domains in the core weighted index having few items, making each of those items more influential in the overall index score.

Our analysis highlighted the importance of competent, motivated human resources to all aspects of delivering

**Table 3** Per cent agreement and Cohen's kappa coefficient among nine Facility Readiness Indices

| | Core simple | Core weighted | | Core PCA | | Expert simple | | Expert weighted | | Expert PCA | | Maximum simple | | Maximum weighted | |
|---|---|---|---|---|---|---|---|---|---|---|---|---|---|---|---|
| | % agreement | % agreement | Kappa | % agreement | Kappa | % agreement | Kappa | % agreement | Kappa | % agreement | Kappa | % agreement | Kappa | % agreement | Kappa |
| **Haiti** | | | | | | | | | | | | | | | |
| Core simple | | | | | | | | | | | | | | | |
| Core weighted | 67.91 | | | | | | | | | | | | | | |
| Core PCA | 69.96 | 51.48 | 0.27 | | | | | | | | | | | | |
| Expert simple | 79.85 | 65.60 | 0.48 | 74.45 | 0.62 | | | | | | | | | | |
| Expert weighted | 74.97 | 65.60 | 0.48 | 66.37 | 0.50 | 77.41 | 0.62 | | | | | | | | |
| Expert PCA | 71.89 | 53.53 | 0.30 | 86.01 | 0.79 | 81.77 | 0.73 | 67.01 | 0.66 | | | | | | |
| Maximum simple | 77.02 | 62.64 | 0.44 | 63.54 | 0.45 | 75.87 | 0.64 | 68.68 | 0.73 | 68.29 | 0.52 | | | | |
| Maximum weighted | 82.54 | 65.98 | 0.49 | 67.91 | 0.52 | 77.79 | 0.67 | 73.68 | 0.64 | 69.58 | 0.54 | 84.47 | 0.77 | | |
| Maximum PCA | 68.68 | 51.73 | 0.28 | 79.46 | 0.69 | 75.48 | 0.69 | 65.08 | 0.63 | 81.90 | 0.73 | 76.77 | 0.65 | 74.58 | 0.62 |
| **Malawi** | | | | | | | | | | | | | | | |
| Core simple | | | | | | | | | | | | | | | |
| Core weighted | 75.58 | | | | | | | | | | | | | | |
| Core PCA | 82.09 | 65.40 | 0.48 | | | | | | | | | | | | |
| Expert simple | 78.04 | 65.28 | 0.48 | 71.90 | 0.58 | | | | | | | | | | |
| Expert weighted | 71.66 | 66.01 | 0.49 | 66.38 | 0.50 | 74.48 | 0.58 | | | | | | | | |
| Expert PCA | 76.69 | 62.70 | 0.44 | 79.51 | 0.69 | 81.10 | 0.72 | 70.18 | 0.62 | | | | | | |
| Maximum simple | 73.13 | 59.88 | 0.40 | 74.85 | 0.62 | 70.67 | 0.56 | 63.68 | 0.56 | 70.18 | 0.55 | | | | |
| Maximum weighted | 77.42 | 64.91 | 0.47 | 77.55 | 0.66 | 73.50 | 0.66 | 65.03 | 0.60 | 70.31 | 0.55 | 79.51 | 0.69 | | |
| Maximum PCA | 79.02 | 63.19 | 0.45 | 86.63 | 0.80 | 78.40 | 0.80 | 66.75 | 0.68 | 80.86 | 0.71 | 82.58 | 0.74 | 80.61 | 0.71 |
| **Tanzania** | | | | | | | | | | | | | | | |
| Core simple | | | | | | | | | | | | | | | |
| Core weighted | 75.19 | | | | | | | | | | | | | | |
| Core PCA | 88.28 | 66.57 | 0.50 | | | | | | | | | | | | |
| Expert simple | 72.69 | 63.36 | 0.45 | 76.80 | 0.65 | | | | | | | | | | |
| Expert weighted | 70.55 | 63.36 | 0.45 | 73.88 | 0.61 | 80.31 | 0.65 | | | | | | | | |
| Expert PCA | 73.65 | 59.73 | 0.40 | 81.56 | 0.72 | 89.77 | 0.88 | 76.26 | 0.70 | | | | | | |
| Maximum simple | 71.62 | 63.36 | 0.45 | 72.75 | 0.59 | 79.83 | 0.59 | 71.62 | 0.70 | 77.57 | 0.66 | | | | |
| Maximum weighted | 77.33 | 65.91 | 0.49 | 77.51 | 0.66 | 75.01 | 0.66 | 71.92 | 0.63 | 74.30 | 0.61 | 84.47 | 0.77 | | |
| Maximum PCA | 71.45 | 59.55 | 0.39 | 76.74 | 0.65 | 80.55 | 0.65 | 72.93 | 0.71 | 81.86 | 0.73 | 84.06 | 0.76 | 79.00 | 0.69 |

Legend

| <0 | 0–0.20 | 0.21–0.40 | 0.41–0.60 | 0.61–0.80 | 0.81–1.0 |
|---|---|---|---|---|---|
| No agreement/worse than chance | Slight | Fair | Moderate | Good | Near perfect |

PCA, principal components analysis.

**Table 4** Variability and Distribution of Index Scores for the nine Facility Readiness Indices

| | Haiti | | | Malawi | | | Tanzania | | |
|---|---|---|---|---|---|---|---|---|---|
| | CoV | Floor effects | Ceiling effects | CoV | Floor effects | Ceiling effects | CoV | Floor effects | Ceiling effects |
| Core simple | 21.20 | 0.00 | 0.00 | 21.70 | 0.00 | 1.10 | 19.11 | 0.00 | 0.48 |
| Core weighted | 22.67 | 0.00 | 0.00 | 21.45 | 0.00 | 1.10 | 20.83 | 0.00 | 0.48 |
| Core PCA | 34.61 | 0.00 | 0.00 | 32.09 | 0.00 | 1.10 | 21.96 | 0.00 | 0.48 |
| Expert simple | 24.30 | 0.00 | 0.39 | 19.64 | 0.00 | 1.23 | 20.14 | 0.00 | 3.33 |
| Expert weighted | 26.01 | 0.00 | 0.39 | 19.82 | 0.00 | 1.23 | 20.24 | 0.00 | 3.33 |
| Expert PCA | 35.72 | 0.00 | 0.51 | 18.96 | 0.00 | 1.23 | 23.62 | 0.00 | 3.33 |
| Maximum simple | 21.12 | 0.00 | 0.00 | 19.33 | 0.00 | 0.00 | 21.31 | 0.00 | 0.00 |
| Maximum weighted | 24.07 | 0.00 | 0.00 | 18.98 | 0.00 | 0.00 | 21.69 | 0.00 | 0.00 |
| Maximum PCA | 26.02 | 0.00 | 0.00 | 20.67 | 0.00 | 1.10 | 22.73 | 0.00 | 0.36 |

CoV, coefficient of variation; PCA, principal components analysis.

high-quality ANC services. Our approach to defining the core set of items highlighted the lack of metrics for human resources captured by facility surveys, as well as the inconsistency between what inputs are required for provision of care and how facility readiness is often defined. This is especially important considering there is a health workforce crisis globally, which is particularly pronounced in LMICs.[42–44] Many countries are suffering from an absolute shortage of healthcare workers. The health workers they do have are often poorly distributed within a country and many workers are deficient in skill mix and core competencies.[45 46]

This study's findings also suggest that to provide more comprehensive measures of ANC quality some items could be added to SPAs and other facility surveys, particularly items related to the experience of care. While the global community is moving towards more person-centred care, data collected through health facility assessments is generally deficient in experience of care measures.[47 48] This may be due to the lack of validated instruments available for measuring the experience of maternal health care.[49] In addition, measures of patient experience from health facility assessments are limited and may be subject to courtesy bias, which has been found to be particularly problematic for subjective questions regarding items such as treatment by staff and consultation quality, items of interest for measuring experience of care.[50] We, therefore, cannot be certain if these measures are actually representative of people's experiences or if other types of questions would better capture experience of care. However, a number of recent studies have generated validated tools for measuring respectful maternity care; these may well provide a starting point for incorporating improved experience of care metrics in health facility assessments such as the SPA.[51 52]

We also found that the indices differed in terms of their ease of construction and interpretation. The simple additive and weighted additive approaches were straightforward to construct because they required taking averages

of items, while the PCA necessitated more complex analysis. In addition, the simple additive approach was the easiest to interpret since facilities with more items available had a higher quality score. The weighted additive approach was also relatively easy to interpret. However, item weighting whereby each domain receives an equal weight was based on an implicit conceptual framework for quality which has not been formally validated. The PCA approach was perhaps the most difficult to interpret as the PCA score represents the linear combination of variables that explained the most variance in the data. As the loadings of each variable on the first component were used as the weights, and some items were negatively correlated and consequently had negative loadings; having all items present in a facility did not always result in the highest PCA score. In addition, the weights assigned by PCA reflected the variation in the data that was different across countries. As a result, PCA did not produce the same weights across different contexts. As we were working with a common definition for what an ANC visit should include across country contexts, this variability in loadings resulted in concerns regarding the face validity and construct validity of the PCA indices. Overall, the low per cent of total variance explained by the first principal component, diverging items with high loadings between countries, negatively correlated items, and the complexity in creating and interpreting the measure highlights concerns for the use of PCA for creating quality of care indices.[53] Another study focused on comparing summary measures of quality of care for family planning found similar results in terms of the ease of construction and interpretation of similar indices suggesting this finding may be relevant across service areas.[54]

Due the variation in ease of construction and interpretation, different methodologies may be better suited for different purposes. For example, the simple additive approach using the maximum number of items may be easiest for Ministries of Health in LMICs to understand and implement. However, if there is particular interest in

**Table 5** Inclusion of items in an index across a range of frequencies for the Facility Readiness Indices

| | Haiti | | | Malawi | | | Tanzania | | |
|---|---|---|---|---|---|---|---|---|---|
| | No/% items <30% frequency | No/% items <40% frequency | No/% items >90% frequency | No/% items <30% frequency | No/% items <40% frequency | No/% items >90% frequency | No/% items <30% frequency | No/% items <40% frequency | No/% items >90% frequency |
| Core (21 items) | 2 (10) | 4 (19) | 5 (24) | 4 (19) | 7 (33) | 6 (29) | 2 (10) | 2 (10) | 8 (38) |
| Expert (19 items) | 2 (11) | 3 (16) | 4 (21) | 2 (11) | 5 (26) | 7 (37) | 0 (0) | 0 (0) | 7 (37) |
| Maximum (38 items) | 8 (21) | 10 (26) | 5 (13) | 8 (21) | 12 (32) | 8 (21) | 5 (13) | 9 (24) | 9 (24) |

using facility readiness as a proxy for provision of care, it might be conceptually important to ensure that items align well between facility readiness and provision of care, thus making the core set of items a good choice. In addition, while the weighted additive method aims to give equal weights to domains of facility readiness and provision of care, as the contents of each domain are determined by what data are available as opposed to an ideal set of items, this approach implicitly weights certain items more heavily because of the unavailability of data on other items and therefore may not be well-suited depending on the purpose of the index. While different methodologies may be better suited for different purposes, there remains a need for standardised, meaningful, valid measures of quality of care that take into account the variation in ease of construction and interpretation and can be used both by countries and at global level.

We note several limitations of the analysis. First, this analysis was conducted using data from three countries, two in Africa and one in Latin America and the Caribbean, which may limit generalisability of findings to low-income countries globally. However, the findings were relatively consistent across the three countries. The two African countries are in the Southern and Eastern Africa region and represent two different types and sizes of health systems within that region. Second, the SPA survey, while quite comprehensive, does not capture every aspect of quality of care that may be important to include in quality of care metrics. However, the SPA represents the most comprehensive information on quality of care in LMICs that is currently available.[34] Third, our approaches to item selection and combination were based on the most commonly used approaches in the literature. However, there are other methods that may be helpful in determining which items to include in a quality of care index, such as latent class analysis, that were not implemented. The selection of methods for this analysis included those most likely to be useful to LMICs. Finally, this analysis does not provide a formal validation of any one ANC quality of care index. However, it does characterise various summary indices of ANC service quality and provides valuable information on the similarities and differences between indices.

## CONCLUSION

While the goal of this study was not to identify the best index for measuring quality of care, we did endeavour to characterise the various indices to make more information available in order to assist health systems researchers in choosing a methodology for creating quality of care indices. Overall, we found the indices to be quite similar within a country. In addition, we found that different methodologies may be better suited for different purposes. Future research on the association between facility readiness and provision of care would be helpful to further characterise the quality of care indices and inform selection of an index. As quality of care becomes

integral to global efforts to improve health outcomes, demand for guidance on creating standardised measures of service quality will grow. This study provides health systems researchers with a comparison of methodologies commonly used to create summary indices of ANC service quality and highlights the similarities and differences between methods. Further research will be required at global and country level to develop standardised, meaningful, valid measures of quality of care that take into account multiple services and various country contexts.

**Acknowledgements** The authors wish to acknowledge the Bill & Melinda Gates Foundation for their support of this project. In addition, we would like to thank the maternal health experts who participated in the expert survey; we appreciate your time and willingness to share your expertise.

**Contributors** AS, MKM, RH and SZ contributed to conceptualising the paper and analysis. AS prepared the manuscript. MKM, RH and SZ critically reviewed and revised the manuscript.

**Funding** This work was supported by the Improving Measurement and Program Design grant (OPP1172551) from the Bill & Melinda Gates Foundation.

**Competing interests** MKM reports grants from Bill and Melinda Gates Foundation during the conduct of the study.

**Patient consent for publication** Not required.

**Ethics approval** The institutional review board at the Johns Hopkins Bloomberg School of Public Health determined this analysis to be exempt from human subjects review.

**Provenance and peer review** Not commissioned; externally peer reviewed.

**Data availability statement** Data are available in a public, open access repository.

**ORCID iD**
Ashley Sheffel http://orcid.org/0000-0001-6561-7085

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
