## [Reviewer comments · BMJ Open]

ARTICLE DETAILS

TITLE (PROVISIONAL)	Development of Summary Indices of Antenatal Care Service Quality in Haiti, Malawi, and Tanzania
AUTHORS	Sheffel, Ashley; Zeger, Scott; Heidkamp, Rebecca; Munos, Melinda

VERSION 1 – REVIEW

REVIEWER	Hannah Leslie Harvard TH Chan School of Public Health
REVIEW RETURNED	15-Jul-2019

GENERAL COMMENTS	Hannah Leslie Harvard TH Chan School of Public Health I appreciate the opportunity to review your manuscript, Development of Summary Indices of Antenatal Care Service Quality in Haiti, Malawi, and Tanzania. This is an important area of study and a timely contribution to the evolving discussion on quality measurement. At present, the manuscript makes clear that the analytic choices in defining and summarizing quality indicators have (in this case relatively modest) implications for the ultimate levels of ‘quality’ and the classification of individual facilities. While useful, especially considered in conjunction with the very similar analysis on family planning that was recently published (see notes in attachment), this conclusion is to be expected and in and of itself does not substantially advance our understanding of quality measurement. I would suggest a more clear engagement with questions of the purpose of quality measurement and the conceptual or theoretical ideas underpinning analytic decisions to provide a better framework for interpreting the findings. In addition, the effort in developing the metrics themselves could be more thoroughly presented to build on this strength of the study. Please find attached more detailed comments on the current version for consideration as you revise. Major • The article currently lacks a discussion of the purpose of these measures or the conceptual basis for measurement, which leaves the assessment of the metrics unmoored and difficult to interpret. For instance, it is not evident what the coefficient of variation would be for a gold-standard measure, which may depend on if the purpose of the measure is to rank all facilities or to identify the worst performing facilities. The authors indicate that it is preferable to have items with low frequency; this would suggest that they are interested in a measure that minimizes measurement error at the high end of quality rather than the low end, but the manuscript does not state this explicitly or justify it either in terms of
---

interpretation or measure performance. The use of floor and ceiling effects as criteria for the measures suggests an interest in dividing facilities / care visits all along a spectrum of quality rather than for instance identifying best and worst performers, but again this is not clearly indicated. Without presenting the purpose of the measure, it is hard to understand the expected outcome of these assessments and the implications of divergence across the analytic methods that are compared. The authors assert that these methods are most likely to be useful in LMIC but it is not clear why – perhaps they are simply the most commonly used?

- Continuing from the first point, regarding the conceptual basis for measurement, the article does not establish a theory for the quality measures shown, which strongly influences how the performance of measures should be judged and what methods may be relevant to compare. If quality is a latent construct measured by reflective indicators, then the correlation of indicators to each other (as measured by Cronbach's alpha for example) is an important indication of good measurement. Criteria such as the distribution of measures across the level of the latent trait (what I think is meant by range of variability in this analysis) come into play. If instead the quality domains measured here are formative constructs composed of the specific, non-interchangeable indicators, then correlation within items is not relevant and distribution of items may be useful only inasmuch as there is an interest in data reduction for measurement efficiency. The choice of measurement methods such as simple additive, weighted additive, and PCA all suggest to some extent a belief in a formative model. The article would be substantially strengthened if the authors are able to provide a more clear framework for these analytic approaches and identify how they do or do not adhere to the goals of a given analysis. Finally, use of the first component from PCA requires a strong assumption of unidimensionality that seems at odds with the domains delineated. Further discussion of the basis for this assumption and the empirical support for it (probably somewhat weak given the PCA results shown, although the remaining Eigenvalues are needed to fully understand this) would be helpful.

- A recent DHS report and subsequent paper applies this exact approach to family planning using the SPA data from the same countries. It would be useful to compare and contrast insights and findings with this work.

<https://journals.plos.org/plosone/article?id=10.1371/journal.pone.0217547>; <https://dhsprogram.com/pubs/pdf/MR20/MR20.pdf>

- Methods:

- o The explanation of CoV is misleading and seems to imply that the measures were compared for how much variance in some other outcome they capture. Preferable to define this as a relative standard deviation to enable comparison of variance accounting for mean level. As noted above, it is not clear at the moment whether a higher or lower CoV is preferable. One could argue that more variation is better to help distinguish facilities from each other, but if variation is driven by inclusion of noisier / more error prone items, reverse would be true.

- o Why is it preferable for indices to have more items that are less frequently available? How is that similar to “items with a range of variability,” which sounds like the item itself should have different variability in different contexts? Are you saying that items across the range of more available to less available should be included (range of variability)? Or that it's better to have items rarely available rather than frequently (consistent low level, unclear what

	the justification for this would be since ceiling effects are assessed already)?  • The methods section is a little hard to follow; it may be easier to define the items (including examples, especially for the HR 'proportion' items), then define the item selection approaches, then the aggregation, and then how these were applied for the readiness and process measures respectively. • The selection of items for indices is emphasized more in the discussion than results, but is an important contribution of this work. It would be helpful to report the alignment of the expert and core items (number of core items not considered important by global experts, expert items not part of the core – this appears in Table 2 but is not presented in the results) and the domains missing in current measurement as part of the results section. Obviously 15 experts don't give a comprehensive assessment, but it does illuminate the challenges in content of current measurement. This would also better support the paragraph in the discussion on competent and motivated HR, which currently feels out of place compared to the results reported in the body of the manuscript • The discussion of weighted additive measures on page 25 is a bit circular – the point of such measures is that domains are pre-defined and equally weighted regardless of the number of items in them, so pointing this out as a limitation is odd. Perhaps the relevant limitation is that there may be items that we know are important but that are undermeasured, so they are currently measured by too few items? Minor  • Abstract does not define the methods used, so mention of them in the results (especially "core weighted") is unclear to the reader; phrase "inclusion of items in an index with a range of variability" is not clear • Bullet points under Strengths and Limitations key points might be stronger as single points, not a weakness with the rebuttal • The introduction mentions child health a bit at random, suggest focusing on maternal or maternal newborn for consistency • Page 10, line 51: "facility ready" should be facility readiness • "Widely used" could be misleading given the small number of countries that have completed SPAs; may be helpful to report the total number of countries/surveys since 2013 before listing the number excluded • Please clarify the inclusion criterion around date of survey – was this motivated by the revision of the SPA survey? Convenience? Change in ANC guidelines or national policy or....? • Were the elements of the complex survey (weights and clustering of observations within facilities) incorporated into the analysis? This is less of a critical element for a measurement analysis than a descriptive analysis, but is still important to clarify whether they were used and if not why not • Were scores actually scaled to be 0 to 100 as stated or were they observed minimum to a possible maximum of 100 as results indicate? • Please report the number of cases that are excluded due to missingness • Figures of the distribution of scores are more valuable than the tables and should be noted more prominently in the results. • SD would be more informative than SE(mean) in Table 1.
--	--

	 • Table 2: please report the number of Eigenvalues over 1 or show these values. As noted above, the use of Cronbach's alpha is at odds with the purpose of PCA (as opposed to factor analysis) and should be justified or reconsidered. Please add a header to indicate these are the loadings on the first component. • How were the studies in S Table 1 identified? Are these supposed to be illustrative of the larger body of literature? It would be helpful to provide a brief description of the search or selection strategy along with this table, particularly if the goal is to be systematic. In case it is helpful, from my own research group I can identify the following articles that include indices of antenatal care readiness and/or process:  o Leslie, H. H., et al. (2016). "Training And Supervision Did Not Meaningfully Improve Quality Of Care For Pregnant Women Or Sick Children In Sub-Saharan Africa." Health Aff (Millwood) 35(9): 1716-1724. o Kruk, M., et al. (2018). "High quality health systems—time for a revolution: Report of the Lancet Global Health Commission on High Quality Health Systems in the SDG Era." Lancet Global Health. o Leslie, H. H., et al. (2017). "Association between infrastructure and observed quality of care in 4 healthcare services: A cross-sectional study of 4,300 facilities in 8 countries." PLoS Med 14(12): e1002464. o Leslie, H. H., et al. (2017). "Effective coverage of primary care services in eight high-mortality countries." BMJ Global Health 2(3). • Why would one assert as the discussion section does that Malawi is likely to be representative of Southern Africa, a region with disparate countries and corresponding health systems?
--	--

REVIEWER	Vanessa Brizuela World Health Organization, Switzerland
REVIEW RETURNED	24-Jul-2019

GENERAL COMMENTS	Development of Summary Indices of Antenatal Care Service Quality in Haiti, Malawi, and Tanzania Thanks for providing me the opportunity to review this manuscript. The authors describe the process for developing indices on quality of antenatal care in three countries using the SPA facility assessment tool as input for data and expert input, together with information from SPA and the FANC for development of the index. The study has some important results relating to standardizing ways in which to measure quality of care using a standardized tool. While I think the overall findings are interesting and important I think there are a few critical changes necessary, especially in the methods section, to ensure publication. Why did you use the WHO Focused Antenatal Care model (from 2002; these are not WHO guidelines) and not the more recent WHO recommendations on antenatal care for a positive pregnancy experience (WHO, 2016. https://www.who.int/reproductivehealth/publications/maternal_perinatal_health/anc-positive-pregnancy-experience/en/)? I would also like to suggest a few papers that might inform the discussion and introduction:
---

Tunçalp Ö, Were WM, MacLennan C, Oladapo OT, Gülmezoglu AM, Bahl R, Daelmans B, Mathai M, Say L, Kristensen F, Temmerman M. Quality of care for pregnant women and newborns—the WHO vision. *BJOG: an international journal of obstetrics & gynaecology*. 2015 Jul 1;122(8):1045-9.

Downe S, Finlayson K, Tunçalp Ö, Gülmezoglu AM. Provision and uptake of routine antenatal services: a qualitative evidence synthesis. *Cochrane Database of Systematic Reviews* 2019, Issue 6. Art. No.: CD012392. DOI: 10.1002/14651858.CD012392.pub2.

Moxon SG, Guenther T, Gabrysch S, Enweronu-Laryea C, Ram PK, Niermeyer S, Kerber K, Tann CJ, Russell N, Kak L, Bailey P. Service readiness for inpatient care of small and sick newborns: what do we need and what can we measure now?. *Journal of global health*. 2018 Jun;8(1).

Madaj B, Smith H, Mathai M, Roos N, van den Broek N. Developing global indicators for quality of maternal and newborn care: a feasibility assessment. *Bulletin of the World Health Organization*. 2017 Jun 1;95(6):445.

Brizuela V, Leslie HH, Sharma J, Langer A, Tunçalp Ö. Measuring quality of care for all women and newborns: how do we know if we are doing it right? A review of facility assessment tools. *The Lancet Global Health*. 2019 May 1;7(5):e624-32.

Bohren MA, Vogel JP, Fawole B, Maya ET, Maung TM, Baldé MD, Oyeniran AA, Ogunlade M, Adu-Bonsaffoh K, Mon NO, Bangoura A. Methodological development of tools to measure how women are treated during facility-based childbirth in four countries: labor observation and community survey. *BMC medical research methodology*. 2018 Dec;18(1):132.

WHO. Quality, equity, dignity: the network to improve quality of care for maternal, newborn and child health – strategic objectives. 2018. Available at: <http://apps.who.int/iris/bitstream/handle/10665/272612/9789241513951-eng.pdf>

Please see specific comments below:

Introduction:

1- Is the list included in Supplementary Table 1 exhaustive? If so, please describe how these were identified. If not, then suggest including them as examples and reference them in the main text and eliminate the table.

2- Also relating to this paragraph, I would limit the description of the methods used in the other studies described.

Methods:

This section is very long and at times a bit confusing. From my understanding you developed a survey using data from SPA, SARA, the FANC model, and expert opinion. You then used the results from this survey together with information collected through the SPA and the framework from WHO to develop an index. You compared data from Haiti, Malawi, and Tanzania against the indices created to determine performance of each index by comparing them with each other. No validation of the surveys was done. Perhaps what you need is a short paragraph explaining the

overall process at the beginning and then a brief description of the steps following that. But it needs to be tightened to ensure conciseness and clarity. This would also help with the existing sub-headers that have no preamble and are hard to understand (e.g., Data, Analysis). More specific comments follow.

SPA

3- I don't think most of this section is necessary as this tool was not developed by the authors for this study. The information included in paragraph 2 starting with "Briefly..." could be included in a table in an appendix.

4- In the second paragraph, suggest replacing "in the last five years..." with "between 2013 and 2018".

5- You state how many surveys there are for each of the observations that you used as inclusion criteria but not how many SPA surveys there were in total. This becomes a bit confusing for the reader. Please add for all or remove.

Maternal health expert survey

6- Please explain how the experts were selected, why you chose to select 50 and what method you used to stop seeking for more responses at 30%.

7- Please use the most updated information re: ANC guidance (see first comment); the one you include are not WHO guidelines and they are from 2002. If the most current recommendations were not used purposely, then I suggest including a justification.

8- Please include the survey as an appendix as this was one of the tools developed for this study.

9- I am unclear about the number 121 items –the WHO standards include 8 standards, 31 quality statements and 352 measures. Where does this 121 derive from (and similarly for the other dimensions you include)?

Analysis

10- You repeat that you limited your analysis to facilities offering ANC –this was described under the header SPA. I'm unsure if you are referring to something different.

Facility readiness and provision of care indices

11- Table numbering should be sequential. You present supplementary table 4 before 3 (page 12)

12- How did you decide to select the 21 items for the core set readiness index?

13- What did you do about the items that were identified by experts as essential but were not included in the SPA questionnaire and in your survey? Did you include them in your index? If you did, please clarify. If you did not, please justify this decision. You mention this again in your discussion section.

14- For the maximum set you mention deciding upon 38 items yet for the core set you mention there was a total of 49 possible items. I am confused as to the selection and exclusion of certain items.

15- You say that you chose expert survey as the only method for the development of the "provision of care" index but I am not convinced by your argument. How are they the best source?

Item combination

As I was reading this section I wasn't convinced of the inclusion of the PCA methodology as it seems complicated and difficult to interpret. And this ends up being your conclusion. This made me

	think that your paper would be strengthened with a more robust justification of using this methodology. Statistical analysis 16- Suggest removing the description of interpretation of Cohen's kappa ranges. I'd limit this to "<0 no agreement to 1 perfect agreement" 17- Please add a justification as to why you wanted to see the variation in the indices. Would this provide you with information about the validity of each index? Similarly, towards the end of paragraph 2 you also mention that having more items that were less frequently available was preferable. 18- I would also like some justification as to the cut-off points used for availability of items (at <30%, <40% and >90%). Results 19- I am unclear why there is a specific sub-section for PCA but not for the other methods 20- I also suggest summarizing the findings to avoid duplication between text and tables/figures Tables & figures Tables are a little busy and the way the data are presented could be improved. The figures were missing titles and are small and hard to read. The colour codes used in the figures are not explained. Table 1. suggest reorganizing the data presented to improve readability Table 2. suggest moving to appendix Table 3. suggest presenting Kappa coefficient and % in same cell (maybe one above the other) since colour codes are linked to % as well. Table 4. can this table be combined with another to reduce the number of tables? Or perhaps included as an appendix? Discussion 21- In the third paragraph you present items that were missing from survey and SPA datasets –this information is more suited for results. In addition, though, why was this information excluded from your survey? These are components of the ANC Guidelines. 22- In the second paragraph, suggest including more recent literature relating to human resource shortages including more recent recommendations for task shifting (reference 24). 23- Suggest including some of references suggested above to support statement with reference 27, and the development of tools to measure experience of care during childbirth by Bohren and colleagues (2018) to support reference 28. 24- Paragraph 4 should include more interaction with the literature –since you say simple additive and weighted additive as well as PCA were used in constructing indices in the past, then perhaps these examples provide some evidence to support or refute your findings. 25- In the middle of this paragraph you state “However, item weighting whereby each domain receives an equal weight was based on a conceptual framework for quality which has not been formally validated.” Which framework are you referring to here? Please include a reference. 26- Fifth paragraph, suggest naming the region “Latin America and the Caribbean”
--	---

	27- You state "...the SPA represents the most comprehensive information on quality of care in LMICs that is currently available." This needs a reference. Conclusions 28- Suggest removing examples from the conclusion as well as description of what the weighted additive methods entail. This correspond more to the discussion and/or methods sections.
--	---

VERSION 1 – AUTHOR RESPONSE

Reviewer 1:

Dear authors,

I appreciate the opportunity to review your manuscript, Development of Summary Indices of Antenatal Care Service Quality in Haiti, Malawi, and Tanzania. This is an important area of study and a timely contribution to the evolving discussion on quality measurement. At present, the manuscript makes clear that the analytic choices in defining and summarizing quality indicators have (in this case relatively modest) implications for the ultimate levels of 'quality' and the classification of individual facilities. While useful, especially considered in conjunction with the very similar analysis on family planning that was recently published (see notes in attachment), this conclusion is to be expected and in and of itself does not substantially advance our understanding of quality measurement. I would suggest a more clear engagement with questions of the purpose of quality measurement and the conceptual or theoretical ideas underpinning analytic decisions to provide a better framework for interpreting the findings. In addition, the effort in developing the metrics themselves could be more thoroughly presented to build on this strength of the study. Please find attached more detailed comments on the current version for consideration as you revise.

Major

- The article currently lacks a discussion of the purpose of these measures or the conceptual basis for measurement, which leaves the assessment of the metrics unmoored and difficult to interpret. For instance, it is not evident what the coefficient of variation would be for a gold-standard measure, which may depend on if the purpose of the measure is to rank all facilities or to identify the worst performing facilities. The authors indicate that it is preferable to have items with low frequency; this would suggest that they are interested in a measure that minimizes measurement error at the high end of quality rather than the low end, but the manuscript does not state this explicitly or justify it either in terms of interpretation or measure performance. The use of floor and ceiling effects as criteria for the measures suggests an interest in dividing facilities / care visits all along a spectrum of quality rather than for instance identifying best and worst performers, but again this is not clearly indicated. Without presenting the purpose of the measure, it is hard to understand the expected outcome of these assessments and the implications of divergence across the analytic methods that are compared. The authors assert that these methods are most likely to be useful in LMIC but it is not clear why – perhaps they are simply the most commonly used?

We have added information in the introduction to clarify what the purpose of the QoC measures is. See page 6,7. In addition, we have added information in the methods section on the purpose of the measures on p.22. A number of different papers and reports have proposed various measures for Qoc. While QoC is often discussed as a latent variable, many of these measures treat it more as a composite or index. We were interested in characterizing these different composite indicators/indices and how they may be similar or different. As such, we aren't proposing a specific measurement model nor choosing methods of comparison based on that measurement model. For our analysis, what we were trying to understand is the usability of these composites/indices by countries and the extent by which they allow researchers/analysts to discriminate between different types of facilities. We aren't

trying to validate a particular measure, but more to characterize the similarities and differences in how facilities may be classified as high or low quality.

- Continuing from the first point, regarding the conceptual basis for measurement, the article does not establish a theory for the quality measures shown, which strongly influences how the performance of measures should be judged and what methods may be relevant to compare. If quality is a latent construct measured by reflective indicators, then the correlation of indicators to each other (as measured by Cronbach's alpha for example) is an important indication of good measurement. Criteria such as the distribution of measures across the level of the latent trait (what I think is meant by range of variability in this analysis) come into play. If instead the quality domains measured here are formative constructs composed of the specific, non-interchangeable indicators, then correlation within items is not relevant and distribution of items may be useful only inasmuch as there is an interest in data reduction for measurement efficiency. The choice of measurement methods such as simple additive, weighted additive, and PCA all suggest to some extent a belief in a formative model. The article would be substantially strengthened if the authors are able to provide a more clear framework for these analytic approaches and identify how they do or do not adhere to the goals of a given analysis. Finally, use of the first component from PCA requires a strong assumption of unidimensionality that seems at odds with the domains delineated. Further discussion of the basis for this assumption and the empirical support for it (probably somewhat weak given the PCA results shown, although the remaining Eigenvalues are needed to fully understand this) would be helpful. We have added information in the introduction to further explain our understanding of these measures as composites or indices as opposed to a latent construct. We have also added information on the purpose of the PCA method on p. 22 "Since the PCA method was used to create weights for a composite variable rather than estimating a latent variable we did not report out the number of components with an eigenvalue greater than one." The Cronbach's alpha has been removed as we agree we shouldn't be assessing reliability of an index (not a latent construct).

- A recent DHS report and subsequent paper applies this exact approach to family planning using the SPA data from the same countries. It would be useful to compare and contrast insights and findings with this work. <https://journals.plos.org/plosone/article?id=10.1371/journal.pone.0217547>; <https://dhsprogram.com/pubs/pdf/MR20/MR20.pdf>
We have added this to the discussion.

Methods

- The explanation of CoV is misleading and seems to imply that the measures were compared for how much variance in some other outcome they capture. Preferable to define this as a relative standard deviation to enable comparison of variance accounting for mean level. As noted above, it is not clear at the moment whether a higher or lower CoV is preferable. One could argue that more variation is better to help distinguish facilities from each other, but if variation is driven by inclusion of noisier / more error prone items, reverse would be true. We have added to the methods section on page 22 more information on why these measures were selected.

- Why is it preferable for indices to have more items that are less frequently available? How is that similar to "items with a range of variability," which sounds like the item itself should have different variability in different contexts? Are you saying that items across the range of more available to less available should be included (range of variability)? Or that it's better to have items rarely available rather than frequently (consistent low level, unclear what the justification for this would be since ceiling effects are assessed already)?

This statement has been removed from the manuscript.

- The methods section is a little hard to follow; it may be easier to define the items (including examples, especially for the HR 'proportion' items), then define the item selection approaches, then the aggregation, and then how these were applied for the readiness and process measures respectively.

The methods section has been reorganized based on this suggestion.

- The selection of items for indices is emphasized more in the discussion than results, but is an important contribution of this work. It would be helpful to report the alignment of the expert and core items (number of core items not considered important by global experts, expert items not part of the core – this appears in Table 2 but is not presented in the results) and the domains missing in current measurement as part of the results section. Obviously 15 experts don't give a comprehensive assessment, but it does illuminate the challenges in content of current measurement. This would also better support the paragraph in the discussion on competent and motivated HR, which currently feels out of place compared to the results reported in the body of the manuscript

We have added a section on the results from the expert survey on p.24.

- The discussion of weighted additive measures on page 25 is a bit circular – the point of such measures is that domains are pre-defined and equally weighted regardless of the number of items in them, so pointing this out as a limitation is odd. Perhaps the relevant limitation is that there may be items that we know are important but that are undermeasured, so they are currently measured by too few items?

This sentence has been edited and moved into the discussion.

Minor

- Abstract does not define the methods used, so mention of them in the results (especially "core weighted") is unclear to the reader; phrase "inclusion of items in an index with a range of variability" is not clear

We have edited the abstract to address this.

- Bullet points under Strengths and Limitations key points might be stronger as single points, not a weakness with the rebuttal

This edit has been made on page 5.

- The introduction mentions child health a bit at random, suggest focusing on maternal or maternal newborn for consistency

The mention of children has been removed.

- Page 10, line 51: "facility ready" should be facility readiness

This edit has been made on page 10.

- "Widely used" could be misleading given the small number of countries that have completed SPAs; may be helpful to report the total number of countries/surveys since 2013 before listing the number excluded

Widely has been removed on page 9. In addition, information has been added on the number of surveys excluded and the reasons for exclusion on page 9.

- Please clarify the inclusion criterion around date of survey – was this motivated by the revision of the SPA survey? Convenience? Change in ANC guidelines or national policy or....?

We have added more detail on how the surveys were selected on p10.

- Were the elements of the complex survey (weights and clustering of observations within facilities) incorporated into the analysis? This is less of a critical element for a measurement analysis than a descriptive analysis, but is still important to clarify whether they were used and if not why not

The elements of the complex survey design (weights and clustering of observation) were not incorporated into the analysis as the goal of this analysis was not to make inferences about the entire population from which the sample of health facilities was drawn. This has been stated on page 16.

- Were scores actually scaled to be 0 to 100 as stated or were they observed minimum to a possible maximum of 100 as results indicate?

We have restated this to say, "We transformed the index into a score out of 100 by dividing by the number of items and multiplying by 100."

- Please report the number of cases that are excluded due to missingness

This has been added in the results on page 18. A total of 6 ANC clients in Haiti, 44 ANC clients in Malawi, and 73 ANC clients in Tanzania were excluded from the analysis due to missing data.

- Figures of the distribution of scores are more valuable than the tables and should be noted more prominently in the results.

We have referenced the figures in the results text and hope that once the article is laid out for publication the figures will appear more prominently.

- SD would be more informative than SE(mean) in Table 1.

We have made this change in table 1.

- Table 2: please report the number of Eigenvalues over 1 or show these values. As noted above, the use of Cronbach's alpha is at odds with the purpose of PCA (as opposed to factor analysis) and should be justified or reconsidered. Please add a header to indicate these are the loadings on the first component.

Since the PCA method was used to create weights for a composite variable rather than estimating a latent variable we did not report out the number of components with an eigenvalue greater than one. We have also removed Cronbach's alpha from the results tables.

- How were the studies in S Table 1 identified? Are these supposed to be illustrative of the larger body of literature? It would be helpful to provide a brief description of the search or selection strategy along with this table, particularly if the goal is to be systematic. In case it is helpful, from my own research group I can identify the following articles that include indices of antenatal care readiness and/or process:

o Leslie, H. H., et al. (2016). "Training And Supervision Did Not Meaningfully Improve Quality Of Care For Pregnant Women Or Sick Children In Sub-Saharan Africa." *Health Aff (Millwood)* 35(9): 1716-1724.

o Kruk, M., et al. (2018). "High quality health systems—time for a revolution: Report of the Lancet Global Health Commission on High Quality Health Systems in the SDG Era." *Lancet Global Health*.

o Leslie, H. H., et al. (2017). "Association between infrastructure and observed quality of care in 4 healthcare services: A cross-sectional study of 4,300 facilities in 8 countries." *PLoS Med* 14(12): e1002464.

o Leslie, H. H., et al. (2017). "Effective coverage of primary care services in eight high-mortality countries." *BMJ Global Health* 2(3).

The studies identified in S Table 1 were meant to be illustrative of the larger body of literature and highlight the differences noted in how summary measures of quality of care have been created. These were extracted from a previous scoping review of the literature aimed at understanding the scope of contemporary SPA and SARA data applications to assess maternal and newborn QoC in LMICs. We have now cited this article and explained these papers are for illustrative purposes (see page 7). As there are many papers published using SPA data to create QoC measures, it's not possible to cite them all individually.

- Why would one assert as the discussion section does that Malawi is likely to be representative of Southern Africa, a region with disparate countries and corresponding health systems?

This statement has been edited on p.31

Reviewer 2:

Thanks for providing me the opportunity to review this manuscript. The authors describe the process for developing indices on quality of antenatal care in three countries using the SPA facility assessment tool as input for data and expert input, together with information from SPA and the FANC for development of the index. The study has some important results relating to standardizing ways in which to measure quality of care using a standardized tool. While I think the overall findings are interesting and important, I think there are a few critical changes necessary, especially in the methods section, to ensure publication.

Why did you use the WHO Focused Antenatal Care model (from 2002; these are not WHO guidelines) and not the more recent WHO recommendations on antenatal care for a positive pregnancy experience (WHO, 2016).

https://www.who.int/reproductivehealth/publications/maternal_perinatal_health/anc-positive-pregnancy-experience/en/?

We used both. The SPA questionnaires were derived from the FANC model and this were the most closely linked to the data available for this analysis. Also, all data collection was completed before the release of the new guidelines. We did however review the new recommendations when developing the survey and this has now been noted on page 10 and 12.

I would also like to suggest a few papers that might inform the discussion and introduction:

Tunçalp Ö, Were WM, MacLennan C, Oladapo OT, Gülmezoglu AM, Bahl R, Daelmans B, Mathai M, Say L, Kristensen F, Temmerman M. Quality of care for pregnant women and newborns—the WHO vision. *BJOG: an international journal of obstetrics & gynaecology*. 2015 Jul 1;122(8):1045-9.

Downe S, Finlayson K, Tunçalp Ö, Gülmezoglu AM. Provision and uptake of routine antenatal services: a qualitative evidence synthesis. *Cochrane Database of Systematic Reviews* 2019, Issue 6. Art. No.: CD012392. DOI: 10.1002/14651858.CD012392.pub2.

Moxon SG, Guenther T, Gabrysch S, Enweronu-Laryea C, Ram PK, Niermeyer S, Kerber K, Tann CJ, Russell N, Kak L, Bailey P. Service readiness for inpatient care of small and sick newborns: what do we need and what can we measure now?. *Journal of global health*. 2018 Jun;8(1).

Madaj B, Smith H, Mathai M, Roos N, van den Broek N. Developing global indicators for quality of maternal and newborn care: a feasibility assessment. *Bulletin of the World Health Organization*. 2017 Jun 1;95(6):445.

Brizuela V, Leslie HH, Sharma J, Langer A, Tunçalp Ö. Measuring quality of care for all women and newborns: how do we know if we are doing it right? A review of facility assessment tools. *The Lancet Global Health*. 2019 May 1;7(5):e624-32.

Bohren MA, Vogel JP, Fawole B, Maya ET, Maung TM, Baldé MD, Oyeniran AA, Ogunlade M, Adu-Bonsaffoh K, Mon NO, Bangoura A. Methodological development of tools to measure how women are treated during facility-based childbirth in four countries: labor observation and community survey. *BMC medical research methodology*. 2018 Dec;18(1):132.

WHO. Quality, equity, dignity: the network to improve quality of care for maternal, newborn and child health – strategic objectives. 2018. Available at:

<http://apps.who.int/iris/bitstream/handle/10665/272612/9789241513951-eng.pdf>

Many thanks for the suggested papers. Some of these papers have been referenced.

Please see specific comments below:

Introduction:

1- Is the list included in Supplementary Table 1 exhaustive? If so, please describe how these were identified. If not, then suggest including them as examples and reference them in the main text and eliminate the table.

The studies identified in S Table 1 were meant to be illustrative of the larger body of literature and highlight the differences noted in how summary measures of quality of care have been created. These were extracted from a previous scoping review of the literature aimed at understanding the scope of contemporary SPA and SARA data applications to assess maternal and newborn QoC in LMICs. We have now cited this article and explained these papers are for illustrative purposes (see page 7). As there are many papers published using SPA data to create QoC measures, it's not possible to cite them all individually.

2- Also relating to this paragraph, I would limit the description of the methods used in the other studies described.

We think it is important to describe this information as it provides more background to the importance of this study.

Methods:

This section is very long and at times a bit confusing. From my understanding you developed a survey using data from SPA, SARA, the FANC model, and expert opinion. You then used the results from this survey together with information collected through the SPA and the framework from WHO to develop an index. You compared data from Haiti, Malawi, and Tanzania against the indices created to determine performance of each index by comparing them with each other. No validation of the surveys was done. Perhaps what you need is a short paragraph explaining the overall process at the beginning and then a brief description of the steps following that. But it needs to be tightened to ensure conciseness and clarity. This would also help with the existing sub-headers that have no preamble and are hard to understand (e.g., Data, Analysis). More specific comments follow. We have reorganized the methods section and added a section called overall approach to improve the clarity of the methods section (pg.9-17).

SPA

3- I don't think most of this section is necessary as this tool was not developed by the authors for this study. The information included in paragraph 2 starting with "Briefly..." could be included in a table in an appendix.

We think the information is useful for readers so that they don't have to go to a full SPA report to have some understanding of the data source therefore we have decided to leave this section in the paper.

4- In the second paragraph, suggest replacing "in the last five years..." with "between 2013 and 2018".

This edit has been made on page 9.

5- You state how many surveys there are for each of the observations that you used as inclusion criteria but not how many SPA surveys there were in total. This becomes a bit confusing for the reader. Please add for all or remove.

Please see updated paragraph on p.10 which should clarify this issue.

Maternal health expert survey

6- Please explain how the experts were selected, why you chose to select 50 and what method you used to stop seeking for more responses at 30%.

We selected experts based on our knowledge of our work in the field and participation in expert groups. We requested respondents to send the survey to other experts within their organization and it was circulated to others who thought they could provide useful responses. After multiple rounds of follow-up, we received a total of 15 responses. As these responses represented a wide range of institutions, we concluded data collection.

7- Please use the most updated information re: ANC guidance (see first comment); the one you include are not WHO guidelines and they are from 2002. If the most current recommendations were not used purposely, then I suggest including a justification.

We did review both when developing the survey and this has now been noted on page 10 and 12.

8- Please include the survey as an appendix as this was one of the tools developed for this study. The survey has been uploaded as an additional supplemental file.

9- I am unclear about the number 121 items –the WHO standards include 8 standards, 31 quality statements and 352 measures. Where does this 121 derive from (and similarly for the other dimensions you include)?

We reviewed the SARA indicators, WHO Focused Antenatal Care (FANC) guidelines, and WHO recommendations on antenatal care for a positive pregnancy experience, as well as the SPA questionnaire. This is where we identified the 121 items. This is not related to the WHO standards with the 8 standards, 31 quality statements and 352 measures as these relate to care at the time of childbirth and our focus here was on antenatal care.

Analysis

10- You repeat that you limited your analysis to facilities offering ANC –this was described under the header SPA. I'm unsure if you are referring to something different.

This first reference refers to inclusion of surveys (i.e. Haiti 2013, Malawi 2013, Tanzania 2014) while the second instance refers to inclusion of particular facilities within each survey.

Facility readiness and provision of care indices

11- Table numbering should be sequential. You present supplementary table 4 before 3 (page 12) This has been edited on page 12.

12- How did you decide to select the 21 items for the core set readiness index?

This is stated on page 12/13. This set of items was identified by reviewing the provision of care items required for an ANC visit based on the WHO FANC guidelines and the WHO recommendations on antenatal care for a positive pregnancy experience, and by determining the human resources, equipment and supplies, medicines, and diagnostics required to deliver each specific item. In creating the core set of items for facility readiness, we mapped each provision of care item to the facility readiness items required to deliver the specific service component. We found that of the 49 provision of care items, 36 items required only human resources and 13 items required human resources plus equipment, diagnostics, medicines, or basic amenities. The core index did not include standard precautions for infection prevention items because these are not explicitly required for any one provision of care item. A total of 21 items were selected for the core set readiness index.

13- What did you do about the items that were identified by experts as essential but were not included in the SPA questionnaire and in your survey? Did you include them in your index? If you did, please clarify. If you did not, please justify this decision. You mention this again in your discussion section. It was not feasible to include items that were identified by experts as essential but were not included in the SPA questionnaire and in the expert survey in the analysis as data on these items was not

collected (the SPA surveys are secondary data). We noted these items in the paper to draw attention to the fact that the available data collection tools may not be sufficiently comprehensive.

14- For the maximum set you mention deciding upon 38 items yet for the core set you mention there was a total of 49 possible items. I am confused as to the selection and exclusion of certain items. This has been clarified on page 14. Out of the 45 facility readiness items identified for inclusion in the expert survey, a total of 38 items were selected for the maximum set readiness index. Seven items from the expert survey were not included in the maximum set of readiness items as data was not collected in the SPA on these items.

15- You say that you chose expert survey as the only method for the development of the “provision of care” index but I am not convinced by your argument. How are they the best source? We have added the following sentence on p.14 “In addition, the experts selected most items as very important or essential, and therefore it was not appropriate to define a core and maximum set of items.” The expert survey method selected almost all items for which data was available in the SPA making little distinction between an expert and a maximum set of items. In addition, we felt there was no approach for selecting a “core” set of items as each action is necessary for a high-quality ANC service. As such, we felt the expert survey was the best source to create the provision of care summary score.

Item combination

As I was reading this section, I wasn't convinced of the inclusion of the PCA methodology as it seems complicated and difficult to interpret. And this ends up being your conclusion. This made me think that your paper would be strengthened with a more robust justification of using this methodology. We have added an explanation for the methods selected in the introduction.

Statistical analysis

16- Suggest removing the description of interpretation of Cohen's kappa ranges. I'd limit this to “<0 no agreement to 1 perfect agreement”
This has been edited on p.22.

17- Please add a justification as to why you wanted to see the variation in the indices. Would this provide you with information about the validity of each index? Similarly, towards the end of paragraph 2 you also mention that having more items that were less frequently available was preferable. We have removed the statement that having more items that were less frequently available was preferable. The aim of this paper was not to validate any one particular index. We were looking at the ability of each measure to capture variation in facilities. Countries want to be able to compare facilities with each other to understand better and worse performers (even if they are all with the low- or high-quality band). We assume that there is some level of variation in QoC across facilities, thus we are interested in being able to capture that variation. All of these measures are assessing the level of variation captured, not necessarily that an index is “better” because it captures more variability, but to understand what is the level of variability being captured. This has now been added to the methods section on p. 22

18- I would also like some justification as to the cut-off points used for availability of items (at <30%, <40% and >90%).

The goal here was to assess the ability to capture variability. These cut-off points were selected in order to identify items that were generally available in most or few facilities and points to the helpfulness of an item in discriminating between higher and lower quality facilities.

Results

19- I am unclear why there is a specific sub-section for PCA but not for the other methods

This sub-section header has been removed.

20- I also suggest summarizing the findings to avoid duplication between text and tables/figures
We have reviewed and feel the text is a useful interpretation of the tables and figures presented.

Tables & figures

Tables are a little busy and the way the data are presented could be improved. The figures were missing titles and are small and hard to read. The colour codes used in the figures are not explained. The figure titles have now been added to the end of the manuscript. In addition, a legend has been added for figure 2.

Table 1. suggest reorganizing the data presented to improve readability

Table 2. suggest moving to appendix

Table 3. suggest presenting Kappa coefficient and % in same cell (maybe one above the other) since colour codes are linked to % as well.

Table 4. can this table be combined with another to reduce the number of tables? Or perhaps included as an appendix?

We appreciate your suggestions on the tables; however, we don't think the tables need to be edited. We will leave this up to the editor of the journal when they are ready to format the article for publication.

Discussion

21- In the third paragraph, you present items that were missing from survey and SPA datasets –this information is more suited for results. In addition, though, why was this information excluded from your survey? These are components of the ANC Guidelines.

We have added a paragraph on the findings from the expert survey to the results section. We did our best to identify all items to include in the expert survey. However, some items were not identified through our review process which led to their exclusion from the survey. There is not a checklist published with the ANC guidelines to facilitate what exactly should be included in a QoC measure and as such some items perhaps were not identified a prior. However, this does not affect the final analysis as we are limited by what the SPA survey collects data on, and those items were all included in the survey.

22- In the second paragraph, suggest including more recent literature relating to human resource shortages including more recent recommendations for task shifting (reference 24).

I have added two more recent references on page 22.

23- Suggest including some of references suggested above to support statement with reference 27, and the development of tools to measure experience of care during childbirth by Bohren and colleagues (2018) to support reference 28.

I have added an additional reference on page 23.

24- Paragraph 4 should include more interaction with the literature –since you say simple additive and weighted additive as well as PCA were used in constructing indices in the past, then perhaps these examples provide some evidence to support or refute your findings.

This literature has been added to the introduction. Additionally, a recently published paper with similar findings has been added to the discussion section.

25- In the middle of this paragraph you state “However, item weighting whereby each domain receives an equal weight was based on a conceptual framework for quality which has not been formally validated.” Which framework are you referring to here? Please include a reference.

This sentence has been changed to the following: However, item weighting whereby each domain receives an equal weight was based on an implicit conceptual framework for quality which has not been formally validated.

26- Fifth paragraph, suggest naming the region “Latin America and the Caribbean”
This has been edited on page 125.

27- You state “...the SPA represents the most comprehensive information on quality of care in LMICs that is currently available.” This needs a reference.
I have added a reference.

Conclusions

28- Suggest removing examples from the conclusion as well as description of what the weighted additive methods entail. This correspond more to the discussion and/or methods sections.
This has been removed and placed in the discussion section.

VERSION 2 – REVIEW

REVIEWER	Hannah Leslie Harvard TH Chan School of Public Health, USA
REVIEW RETURNED	07-Oct-2019

GENERAL COMMENTS	Thank you for the revised manuscript and your attention to the range of reviewer comments. The revised manuscript is clear and follows the comparative purpose as stated. I have relatively minor comments on technical issues, below. I also include a broader comment on the value of this work purely for consideration. - The decision to position this as a comparative descriptive rather than measurement analysis seems at odds with the decision not to consider weights or clustering. While measurement analyses may not require inclusion of the study design, this work is clear to focus on usability than expected vs actual performance. It thus focuses on the practical performance of measures - including the difference in quantities like the mean and variance - in these specific data. Disregarding the survey design is puzzling in this case since there are implications of survey design for these quantities. It’s possible that the weights and clustering would not change the relative performance of the measures, but I don’t think that is a guarantee. The proportion of facilities with an item could be strongly affected in Tanzania given the oversampling of hospitals and the generally higher readiness in hospitals. The stated reason of not drawing inference does not seem to square with the discussion and recommendations that these findings can inform future decision making - if a decision maker in Tanzania wants to select an index with both rare and common items, isn't the target population of interest all health facilities rather than this sample which is disproportionately hospitals? Is the idea that since not all facilities include antenatal care observations, the sample is no longer generalizable to the full health system and so the weights do not help to recreate the target population of facilities offering ANC? This should not affect the decision about including strata or clustering, and if it is the rationale, should be stated and cited / justified with reference to the study design. It is possible that incorporating the survey design does not make a difference in this
---

	analysis, but the explanation is not convincing given the overall aim of the article to present empirical findings to assist researchers and decision makers in understanding the pragmatic, quantitative trade offs between selected measures in their application to specific data.  - Please cite and define the benchmarks used for kappa agreement (fair, good, etc). - Page 15 seems to be missing the number of items on the expert readiness index - For consideration: the manuscript emphasizes the message that analytic choices have make minimal difference on the ratings of facilities on this particular measure in these three countries, with the implication that decision-makers and researchers should feel free to choose a measure based on ease of calculation and interpretation. Without consideration of the conceptual reasoning behind these approaches or empirical benchmarks for the comparisons, there's fairly little to go on in deciding whether similar conclusions could be drawn on future measures or studies. For instance, because no standard is provided for understanding the values for things like CoV and % items in each category, it is difficult to assess the evidence for the statement that the differences between measures are minor or inconsequential - how would a reader know that a difference in CoV of 14% as is seen between measures in Haiti is small enough that these measures can be considered 'quite similar'? (This conclusion of similarity also seems more sweeping than the results showing differences, such as mean process quality in Malawi.) It does not require a fully fledged measurement theory to recognize that the methods used here reflect distinct beliefs about what is important to measure - specific items, specific domains, or the variance captured by a suite of items. Decision makers may have a preference among these beliefs even, as if in the example here, that preference does not have tremendous implications for resulting facility classification. In emphasizing the empirical results of different measurement approaches, the paper makes a narrow point about these specific measures in these particular surveys rather than bringing broader clarity to the muddled field of quality measurement around why, what, and how one might measure.
--	--

REVIEWER	Vanessa Brizuela World Health Organization, Switzerland
REVIEW RETURNED	15-Oct-2019

GENERAL COMMENTS	Development of Summary Indices of Antenatal Care Service Quality in Haiti, Malawi, and Tanzania Thanks for providing a revised and much improved version of the manuscript. The changes made in the methods section now allow for a much easier flow and understanding. Overall, however, there seem to be a few assumptions made that are not entirely clear. Your assumptions might be correct and the best possible way to proceed with the analysis, but I think these should be made explicit. For example: a- You chose the SPA tool for your analysis and you explain that it is widely used, but this does not necessarily convince the reader as to why. There is also the issue that not all versions of the tool are used in all settings (i.e., the exit interviews are not included in all the surveys). Perhaps a brief explanation that this tool was chosen for this exercise because of its widespread use but that
--

	there are many different tools available (including the SARA which you use for your development of indices) –which might also favour your argument that there is little guidance for decision makers as to what to measure and how. b- You decide to exclude all standard precautions for infection prevention and control from your core set because you assume these are not required for provision of care. One could argue that these are essential items for provision of care (e.g., handwashing). Please explain your rationale. c- You state that health experts are “the best source for determining which processes are essential to high quality of ANC.” This is a big assumption which might be more convincing if the process of selecting these health experts is clearer/more transparent and the assumption is made evident. Linked to the above, perhaps a more precise explanation on how the sample of experts was selected would be good. I am assuming this was a purposive sample of maternal health experts, but it is unclear how they were identified (people you knew? Authors to major publications in the past X number of years? Directors of maternal health programmes... where?). In addition, I was not entirely convinced about why examining the variability and distribution of the indices responds to the goal of your paper to “develop, characterize, and compare” the indices. This is especially important when considering the audience of your paper and recommendations and capacity to interpret your results. I suggest you think about whether this is essential to your message. If so, then a bit more linkage might be needed. If not, figure out if the overall message works (and, this might help in making the manuscript shorter; as is, it is well beyond the 4,000 word proposed length). I include some specific comments below on small discrepancies and issues. Abstract: 1- In the methods section you removed that you calculated floor and ceiling effects but you then report this in the results section. Relatedly, you do report on this in the main manuscript so there is some discrepancy. Introduction: 2- Page 10, reference 16 is a little misleading as the paper cited refers to the use of indices and composite scores, but not specific to the topic at hand. I suggest replacing “these studies often use” with “summary of indices or composite scores are often used in measurement models”, which is closer to what the paper indicates. 3- Page 11. You state “to date, few studies have compared...”. It would be important to reference those few you have identified, and if there already are some studies that have done what you are doing with this paper, state clearly what your paper adds to those previous studies, why your paper is necessary. This last aspect is crucial. Methods: 4- Page 17. You explain how items were identified and then state “... and by determining the human resources, equipment and supplies...”. How was this determination made?
--	---

	Results 5- Page 31. In the second paragraph there seems to be a number missing. You state “The core readiness index included 21 items, the expert readiness index included items...” 6- Page 33. You mention that there were higher IQR using PCA in all three countries, yet in Malawi IQR is lowest for expert PCA (according to results in table 1). 7- Page 35. There is a discrepancy in numbers from table 2 to what is shown here. % variance ranged from 9.89 in Malawi and 10.49 in Haiti, according to numbers in table 2. Also, on page 36 CoV in Haiti for expert PCA was 35.72 (as per table 4). Conclusion 8- Perhaps it would be useful to state here that overall you found a lot of similarities among the three methods and the different indices according to selection of items, in addition to stating that some might be better than others.
--	--

VERSION 2 – AUTHOR RESPONSE

Reviewer 1:

Thank you for the revised manuscript and your attention to the range of reviewer comments. The revised manuscript is clear and follows the comparative purpose as stated. I have relatively minor comments on technical issues, below. I also include a broader comment on the value of this work purely for consideration.

- The decision to position this as a comparative descriptive rather than measurement analysis seems at odds with the decision not to consider weights or clustering. While measurement analyses may not require inclusion of the study design, this work is clear to focus on usability than expected vs actual performance. It thus focuses on the practical performance of measures - including the difference in quantities like the mean and variance - in these specific data. Disregarding the survey design is puzzling in this case since there are implications of survey design for these quantities. It's possible that the weights and clustering would not change the relative performance of the measures, but I don't think that is a guarantee. The proportion of facilities with an item could be strongly affected in Tanzania given the oversampling of hospitals and the generally higher readiness in hospitals. The stated reason of not drawing inference does not seem to square with the discussion and recommendations that these findings can inform future decision making - if a decision maker in Tanzania wants to select an index with both rare and common items, isn't the target population of interest all health facilities rather than this sample which is disproportionately hospitals? Is the idea that since not all facilities include antenatal care observations, the sample is no longer generalizable to the full health system and so the weights do not help to recreate the target population of facilities offering ANC? This should not affect the decision about including strata or clustering, and if it is the rationale, should be stated and cited / justified with reference to the study design. It is possible that incorporating the survey design does not make a difference in this analysis, but the explanation is not convincing given the overall aim of the article to present empirical findings to assist researchers and decision makers in understanding the pragmatic, quantitative trade offs between selected measures in their application to specific data.

Many thanks for your comment on the appropriateness of the use of the survey design for this analysis. We recognize that as we are using the same set of facilities for each index, using the weights would have an equal effect across all measures. In addition, there should be no effect of clustering for the readiness indices. For the provision of care indices, all items are the same in the three indices. Thus, we wouldn't expect to see different design effects across the three indices. The effect of stratification should be negligible. If anything, the SEs for our analysis are conservative (if we had accounted for stratification, we would likely see slightly smaller SE). Finally, we plan to expand this analysis in a second paper to look at the association between readiness and provision of care using regression (without accounting for the survey design) which was an additional consideration in making this analytical choice.

- Please cite and define the benchmarks used for kappa agreement (fair, good, etc).

This has been added back in on p.16. It was previously removed as per the suggestion of other reviewers.

- Page 15 seems to be missing the number of items on the expert readiness index
The omitted number (19) has been added to page 18.

- For consideration: the manuscript emphasizes the message that analytic choices have made minimal difference on the ratings of facilities on this particular measure in these three countries, with the implication that decision-makers and researchers should feel free to choose a measure based on ease of calculation and interpretation. Without consideration of the conceptual reasoning behind these approaches or empirical benchmarks for the comparisons, there's fairly little to go on in deciding whether similar conclusions could be drawn on future measures or studies. For instance, because no standard is provided for understanding the values for things like CoV and % items in each category, it is difficult to assess the evidence for the statement that the differences between measures are minor or inconsequential - how would a reader know that a difference in CoV of 14% as is seen between measures in Haiti is small enough that these measures can be considered 'quite similar'? (This conclusion of similarity also seems more sweeping than the results showing differences, such as mean process quality in Malawi.) It does not require a fully fledged measurement theory to recognize that the methods used here reflect distinct beliefs about what is important to measure - specific items, specific domains, or the variance captured by a suite of items. Decision makers may have a preference among these beliefs even, as if in the example here, that preference does not have tremendous implications for resulting facility classification. In emphasizing the empirical results of different measurement approaches, the paper makes a narrow point about these specific measures in these particular surveys rather than bringing broader clarity to the muddled field of quality measurement around why, what, and how one might measure.

We agree there is a huge lack of clarity and an enormous amount of work to be done to create meaningful and valid measures of structure and process quality. This analysis represents an initial attempt to grapple with this by comparing different analytical approaches that have been used by other researchers to generate readiness and provision of care index scores and to see whether, in these selected settings, the different approaches produce different results. We were surprised to find that the indices weren't all that different in these settings. We've added a sentence to the discussion to clarify we believe there is a need for an agreed upon set of standardized, meaningful, valid measures of quality of care as the proliferation of QoC measures currently used makes it difficult to track progress in service quality. In addition, we've added a sentence to the conclusion to clarify this is a preliminary attempt to engage with the situation of quality measurement and have suggested some further research to be done at global and country level to come up with standardized, meaningful, valid measures of QoC that take into account multiple services and various country contexts.

Reviewer 2:

Thanks for providing a revised and much improved version of the manuscript. The changes made in the methods section now allow for a much easier flow and understanding.

Overall, however, there seem to be a few assumptions made that are not entirely clear. Your assumptions might be correct and the best possible way to proceed with the analysis, but I think these should be made explicit. For example:

- a- You chose the SPA tool for your analysis and you explain that it is widely used, but this does not necessarily convince the reader as to why. There is also the issue that not all versions of the tool are used in all settings (i.e., the exit interviews are not included in all the surveys). Perhaps a brief explanation that this tool was chosen for this exercise because of its widespread use but that there are many different tools available (including the SARA which you use for your development of indices) –which might also favour your argument that there is little guidance for decision makers as to what to measure and how.
We have added a sentence on p.9 to clarify our choice of SPA data for this analysis.

- b- You decide to exclude all standard precautions for infection prevention and control from your core set because you assume these are not required for provision of care. One could argue that these are essential items for provision of care (e.g., handwashing). Please explain your rationale. Ideally, people should be washing their hands before every client. However, for the core set of items we wanted to look closely at items that are specifically related to ANC and are likely related to the provision of care. For this reason, we also left out infrastructure items from the core.
- c- You state that health experts are “the best source for determining which processes are essential to high quality of ANC.” This is a big assumption which might be more convincing if the process of selecting these health experts is clearer/more transparent and the assumption is made evident.

Linked to the above, perhaps a more precise explanation on how the sample of experts was selected would be good. I am assuming this was a purposive sample of maternal health experts, but it is unclear how they were identified (people you knew? Authors to major publications in the past X number of years? Directors of maternal health programmes... where?).

We selected experts based on our knowledge of our work in the field and participation in expert groups (including WHO, UNICEF, IMHM/EPMM, MoNITOR, etc.). We requested respondents to send the survey to other experts within their organization and it was circulated to others who thought they could provide useful responses. After multiple rounds of follow-up, we received a total of 15 responses. As these responses represented a wide range of institutions, we concluded data collection.

In addition, I was not entirely convinced about why examining the variability and distribution of the indices responds to the goal of your paper to “develop, characterize, and compare” the indices. This is especially important when considering the audience of your paper and recommendations and capacity to interpret your results. I suggest you think about whether this is essential to your message. If so, then a bit more linkage might be needed. If not, figure out if the overall message works (and, this might help in making the manuscript shorter; as is, it is well beyond the 4,000 word proposed length). The variability and distribution of the indices are key characteristics of the indices therefore we feel they are central to the message.

I include some specific comments below on small discrepancies and issues.

Abstract:

- 1- In the methods section you removed that you calculated floor and ceiling effects but you then report this in the results section. Relatedly, you do report on this in the main manuscript so there is some discrepancy. Floor and ceiling effects are incorporated under “examining the variability and distribution of scores” in the methods section of the abstract. Due to word limitations we cannot list each measure individually.

Introduction:

- 2- Page 10, reference 16 is a little misleading as the paper cited refers to the use of indices and composite scores, but not specific to the topic at hand. I suggest replacing “these studies often use” with “summary of indices or composite scores are often used in measurement models”, which is closer to what the paper indicates. That paper was intended to support the definition of summary and composite indices. We have added additional references to refer to the studies which have used indices.
- 3- Page 11. You state “to date, few studies have compared...”. It would be important to reference those few you have identified, and if there already are some studies that have

done what you are doing with this paper, state clearly what your paper adds to those previous studies, why your paper is necessary. This last aspect is crucial. This has been edited to read “To our knowledge, no studies have compared...”

Methods:

4- Page 17. You explain how items were identified and then state “... and by determining the human resources, equipment and supplies...”. How was this determination made? We made this determination based on our knowledge of maternal health care.

Results

5- Page 31. In the second paragraph there seems to be a number missing. You state “The core readiness index included 21 items, the expert readiness index included items...” The omitted number (19) has been added to page 18.

6- Page 33. You mention that there were higher IQR using PCA in all three countries, yet in Malawi IQR is lowest for expert PCA (according to results in table 1). The lowest IQR for Malawi is not the expert PCA, it is the maximum weighted index. We have added the statement “in general” to that sentence to clarify.

7- Page 35. There is a discrepancy in numbers from table 2 to what is shown here. % variance ranged from 9.89 in Malawi and 10.49 in Haiti, according to numbers in table 2. Also, on page 36 CoV in Haiti for expert PCA was 35.72 (as per table 4). The % variance explained by the first principal component has been updated as follows: 9.89% in Malawi to 10.49% in Haiti.

The CoV for Haiti has been changed from 35.62 to 35.72.

Conclusion

8- Perhaps it would be useful to state here that overall you found a lot of similarities among the three methods and the different indices according to selection of items, in addition to stating that some might be better than others. We have added a sentence to the conclusion on p.27

VERSION 3 – REVIEW

REVIEWER	Hannah Leslie Harvard TH Chan School of Public Health
REVIEW RETURNED	28-Oct-2019

GENERAL COMMENTS	I appreciate your thoughtful responses to the continuing review. Your points regarding clustering and weighting are well taken, though I am not persuaded that Table 5 on item frequencies would be unaffected by weighting or that any difference here is immaterial to the criterion used in this analysis. I certainly appreciate and agree with the call for better measurement work. I wonder what specific studies you would like to see that you were not able to address in this paper, although perhaps this is a topic for future discussion.
---

REVIEWER	Vanessa Brizuela World Health Organization, Switzerland
REVIEW RETURNED	31-Oct-2019

GENERAL COMMENTS	Thanks for the work put into the paper "Development of Summary Indices of Antenatal Care Service Quality in Haiti, Malawi, and Tanzania" and the revisions made thus far.
---

	I don't have any additional major comments at this time.
--	--